# Transcriptional signatures of schizophrenia in hiPSC-derived NPCs and neurons are concordant with post-mortem adult brains

Gabriel E. Hoffman [1,2], Brigham J. Hartley[3,4], Erin Flaherty[4,5], Ian Ladran[3,4], Peter Gochman[6], Douglas M. Ruderfer[1,2,7], Eli A. Stahl[1,2], Judith Rapoport[6], Pamela Sklar[1,3,4,5] & Kristen J. Brennand [1,2,3,4]

The power of human induced pluripotent stem cell (hiPSC)-based studies to resolve the smaller effects of common variants within the size of cohorts that can be realistically assembled remains uncertain. We identified and accounted for a variety of technical and biological sources of variation in a large case/control schizophrenia (SZ) hiPSC-derived cohort of neural progenitor cells and neurons. Reducing the stochastic effects of the differentiation process by correcting for cell type composition boosted the SZ signal and increased the concordance with post-mortem data sets. We predict a growing convergence between hiPSC and post-mortem studies as both approaches expand to larger cohort sizes. For studies of complex genetic disorders, to maximize the power of hiPSC cohorts currently feasible, in most cases and whenever possible, we recommend expanding the number of individuals even at the expense of the number of replicate hiPSC clones.

[1] Department of Genetics and Genomics, Icahn School of Medicine at Mount Sinai, New York, NY 10029, USA. [2] Icahn Institute of Genomics and Multiscale Biology, Icahn School of Medicine at Mount Sinai, New York, NY 10029, USA. [3] Department of Neuroscience, Icahn School of Medicine at Mount Sinai, New York, NY 10029, USA. [4] Friedman Brain Institute, Icahn School of Medicine at Mount Sinai, New York, NY 10029, USA. [5] Department of Psychiatry, Icahn School of Medicine at Mount Sinai, New York, NY 10029, USA. [6] Childhood Psychiatry Branch, National Institute of Mental Health, National Institutes of Health, Bethesda, MD 20892, USA. [7] Present address: Division of Genetic Medicine, Departments of Medicine, Psychiatry and Biomedical Informatics, Vanderbilt Genetics Institute, Vanderbilt University Medical Center, Nashville, TN 37232, USA. Correspondence and requests for materials should be addressed to G.E.H. (email: gabriel.hoffman@mssm.edu) or to K.J.B. (email: kristen.brennand@mssm.edu)

A growing number of studies have demonstrated that human induced pluripotent stem cells (hiPSCs) can serve as cellular models of both syndromic and idiopathic forms of a variety of neurodevelopmental disorders (reviewed in ref. [1]). We and others have previously shown that hiPSC-derived neural progenitor cells (NPCs) and neurons generated from patients with schizophrenia (SZ) show altered gene expression[2–4], which may underlie observed in vitro phenotypes such as aberrant hiPSC-NPC polarity[5] and migration[6], as well as deficits in hiPSC-neuron connectivity and function[3,7]. Altogether, such hiPSC-based approaches seem to capture aspects of SZ biology identified through post-mortem studies and animal models[8]. Nonetheless, mechanistic studies to date have tended to focus on rare variants[3–5]; the ability of an hiPSC-based approach to resolve the much smaller effects of common variants remained uncertain.

We established a case-control SZ cohort structure designed to capture a broad range of rare and common variants that might underlie SZ risk, in order to address and quantify the intra- and inter-individual variability inherent in this approach and uncover to what extent hiPSC-based models can identify common pathways underlying such different genetic risk factors. Because hiPSC-neurons are likely best suited for the study of disease predisposition[6], we applied this methodology to a childhood-onset SZ (COS) cohort, a subset of SZ patients defined by onset, severity and prognosis. COS patients have a more salient genetic risk, with a higher rate of SZ-associated copy number variants (CNVs)[9] and stronger common SZ polygenic risk scores[10]. Overall, across 94 RNA-Seq samples, we observed many sources of variation reflecting both biological (i.e., reprogramming and differentiation) and technical effects. By systematically accounting for covariates and adjusting for heterogeneity in neural differentiation, we improved our ability to resolve the disease-relevant signal. Our bioinformatic pipeline reduces the risk of false positives arising from the small sample sizes of hiPSC-based approaches and we hope it can help guide data analysis in similar hiPSC-based disease studies.

## Results

**Transcriptomic profiling of COS hiPSC-NPCs and hiPSC-neurons.** Individuals with COS, as well as unaffected, unrelated healthy controls were recruited as part of a longitudinal study conducted at the National Institute of Health[9,10] (see Supplementary Data 1 for available clinical information). This cohort is comprised of nearly equal numbers of cases and controls (Fig. 1a–c); 16 cases were selected representing a range of SZ-relevant CNVs, including 22q11.2 deletion, 16p11.2 duplication, 15q11.2 deletion, and *NRXN1* deletion (2p16.3)[11] and/or idiopathic genetics with a strong family history of SZ, 12 controls were identified as being most appropriately matched for sex, age, and ethnicity (Fig. 1d; Supplementary Data 1).

We used an integration free approach to generate genetically unmanipulated hiPSCs from COS patients (14 of 16 patients, 88% reprogrammed) and unrelated age- and sex-matched controls (12 of 12 controls, 100% reprogrammed) (Fig. 1b). Briefly, primary fibroblasts were reprogrammed by sendai viral delivery of *KLF4*, *OCT4*, *SOX2*, and *cMYC*; presumably clonal lines were picked and expanded 23–30 days following transduction. Following extensive immunohistochemistry, fluorescent activated cell sorting (FACS), quantitative polymerase chain reaction (qPCR) and karyotype assays to assess the quality of the hiPSCs (Fig. 1b, e, f), we selected two to three presumably clonal hiPSC lines per individual ($n = 40$ COS, $n = 35$ control, Table 1; Supplementary Data 1). A subset of these hiPSCs has been previously reported[2].

Using dual-SMAD inhibition, three to five forebrain hiPSC-NPC populations were differentiated from each validated hiPSC

line via an embryoid body intermediate[6], once hiPSCs had been passaged ~10 times. hiPSC-NPCs with normal morphology and robust protein levels of NESTIN and SOX2 by FACS and/or immunocytochemistry (Fig. 1g, h) ($n = 32$ COS, $n = 35$ control hiPSC-NPCs representing 67 unique hiPSC lines reprogrammed from 12 unique COS and 12 unique control individuals) were selected for further differentiation to 6-week-old forebrain neuron populations (Table 1; Supplementary Data 2). We have previously demonstrated that hiPSC-NPCs can be directed to differentiate into mixed populations of excitatory neurons, inhibitory neurons and astrocytes[7]. hiPSC-neurons have neuronal morphology, undergo action potentials, release neurotransmitters, show evidence of spontaneous synaptic activity, and resemble the gene expression of fetal forebrain tissue.

Because it required nearly 4 years to generate and differentiate all hiPSCs, hiPSC-NPCs, and hiPSC-neurons, it was not possible to fully apply standardized conditions across all cellular reprogramming and neural differentiations. Media reagents, substrates, and growth factors for fibroblast expansion, reprogramming, hiPSC differentiation, NPC expansion, and neuronal differentiation, as well as personnel and laboratory spaces, varied over time. Although individual fibroblast lines were reprogrammed and differentiated to hiPSC-NPCs in the order in which they were received, multiple randomization steps were introduced at the subsequent stages, particularly the thaw, expansion, and neuronal differentiation of validated hiPSC-NPCs in preparation for RNA sequencing (RNA-Seq) (see Supplementary Data 2 for available batch information). Only validated hiPSC-NPCs that yielded high quality populations of matched hiPSC-NPCs and hiPSC-neurons in one of three batches of thaws were used for RNA-Seq (Supplementary Data 1, 2).

RNA-Seq data were generated from 94 samples ($n = 47$ hiPSC-NPC, $n = 47$ hiPSC-neurons; $n = 46$ COS, $n = 48$ controls; representing 42 unique hiPSC lines reprogrammed from 11 unique COS and 11 unique control individuals) following ribosomal RNA (rRNA) depletion (Table 1; Supplementary Data 2). The median number of uniquely mapped read pairs per sample was 42.7 million, of which only a very small fraction were rRNA reads (Supplementary Fig. 1; Supplementary Data 3). In total 18,910 genes (based on ENSEMBL v70 annotations) were expressed at levels deemed sufficient for analysis (at least 1 CPM in at least 30% of samples); 11,681 were protein coding, 879 were lncRNA, and the remaining were of various biotypes (Supplementary Data 4).

Since six COS patients were selected based on CNV status, we examined gene expression in the regions affected by the CNVs. Despite the noise inherent to RNA-Seq and the high level of biologically driven expression variation in samples without CNVs, we identified corresponding hiPSC-NPC and hiPSC-neuron expression changes in some CNV regions (Supplementary Fig. 2).

In addition to SZ diagnosis-dependent effects, gene expression between hiPSC-NPCs and hiPSC-neurons was expected to vary as a result of technical[12], epigenetic[13], and genetic[14] differences[15]. Unexpectedly, we also observed substantial variation in cell type composition (CTC) between populations of hiPSC-NPCs and hiPSC-neurons. In the following sections, we discuss our strategy to address these sources of variation.

**Addressing technical variation in RNA-Seq data.** We implemented an extensive quality control pipeline to detect, minimize and account for many possible sources of technical variation (Fig. 1i). Samples were submitted and processed for RNA-Seq in only one batch; RNA isolation, library preparation, and sequencing were completed under standardized conditions at the New

York Genome Center. Errors in sample mislabeling and cell culture contamination were identified, allowing us to correct sample labeling when possible and remove samples from further analysis when not. Batch effects in both tissue culture and RNA-Seq sample processing were corrected for and samples with

aberrant X-inactivation[16] and/or residual Sendai virus expression were flagged.

Expression patterns of genes on the sex chromosomes can identify the sex of each sample, confirm sample identity, and also measure the extent of X-inactivation in females. Using *XIST* on

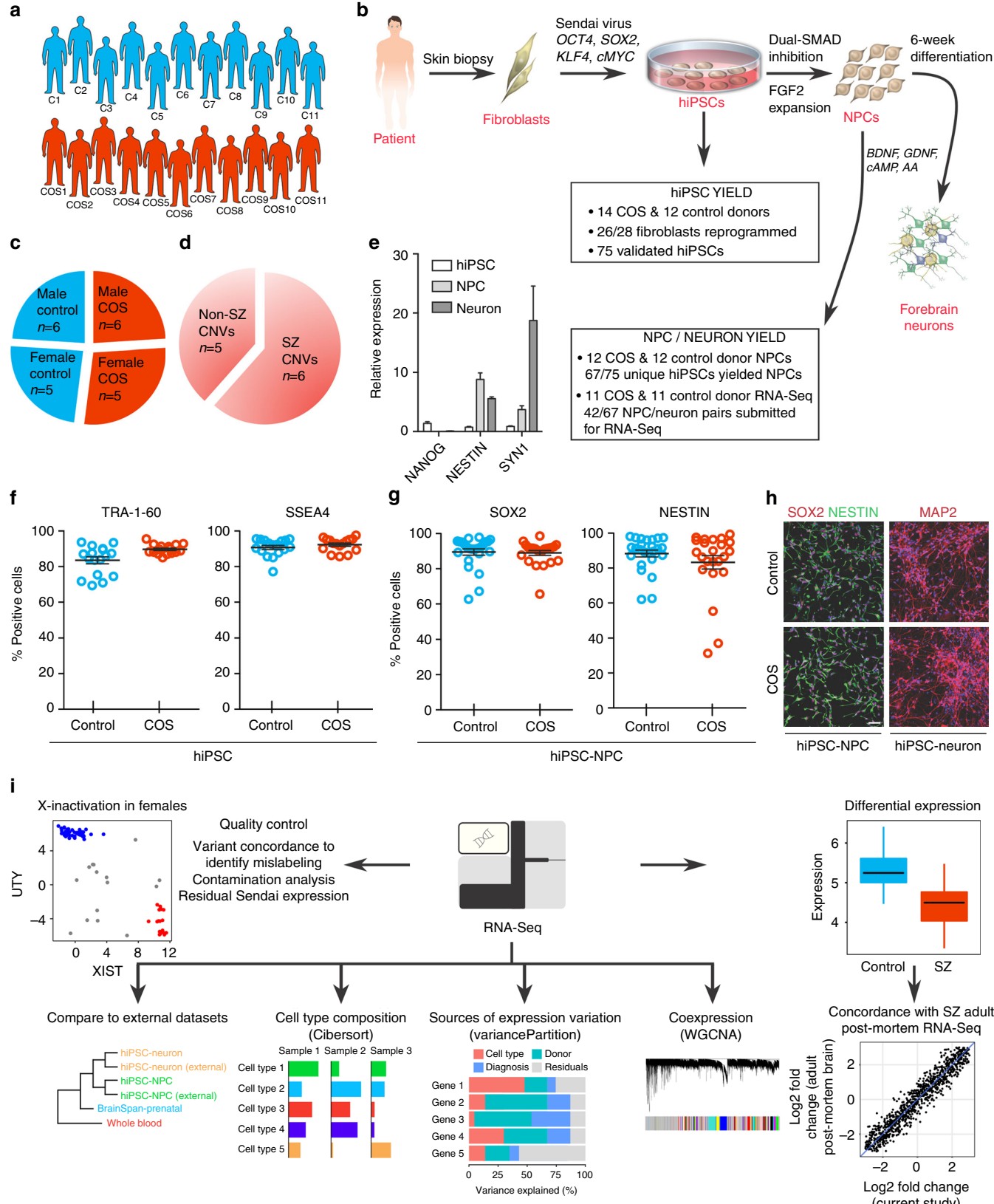

chrX and the expression of six genes on chrY (*USP9Y*, *UTY*, *NLGN4Y*, *ZFY*, *RPS4Y1*, *TXLNG2P*), this analysis identified 2 mislabeled males that show a female expression pattern and 15 female samples that have expression patterns intermediate between males and females (Supplementary Fig. 3A), consistent with either contamination or aberrant X-inactivation.

Samples with mislabeling and/or cross-individual contamination, whether during cell culture and/or RNA library preparation, were identified through genotype concordance analysis. Verify-BamID[17] was used to compare the genotype of the source fibroblast samples with variants called from RNA-Seq data from the respective hiPSC-NPCs and hiPSC-neurons. In total, 76 samples (81%; $n = 38$ hiPSC-NPC, $n = 38$ hiPSC-neurons; $n = 36$ COS, $n = 40$ controls, from 10 unique COS and 9 unique control individuals) were validated for subsequent analysis (Table 1; Supplementary Data 2; Supplementary Fig. 3B).

Residual Sendai virus expression was assessed using Inchworm in the Trinity package[18], which performed de novo assembly of reads that did not map to the human genome. Comparisons of these contigs to the Sendai virus genome sequence (GenBank: AB855655.1) quantified the number of reads corresponding to residual Sendai expression in each NPC and neuron sample. Although Sendai viral vectors are widely assumed to be lost within 11 hiPSC passages[19], and that on average our hiPSCs were passaged > 10–15 times and our hiPSC-NPCs > 5 times, we identified Sendai viral transcripts in a subset of our samples. While the majority (70 of 87, 80%) (75 of the total 94, 79.8%) of RNA-Seq samples did not contain any reads that mapped to the Sendai viral genome, 17 (or 19 of total) samples (Supplementary Data 2; Supplementary Fig. 4) showed evidence of persistent Sendai viral expression at > 1 count per million. Differential expression analysis identified 2768 genes correlated with Sendai expression at FDR < 5% (Supplementary Data 5). We note that this signal is not driven by outliers since quantile normalized Sendai expression values were used in this analysis. In fact, these genes are highly enriched for targets of *MYC* (OR = 3.75, $p < 6.4$e-38) (Supplementary Data 6, Supplementary Fig. 5A). Although *MYC* is one of the four transcription factors (along with *SOX2*, *KLF4*, and *OCT4*) used in hiPSC reprogramming, expression of these four genes was not associated with Sendai expression (Supplementary Fig. 5B). The correlation of residual Sendai expression with activation of *MYC* targets suggests that this could be a potential source of transcriptional and phenotypic variation in hiPSCs; however, neither incorporating Sendai expression as a covariate nor dropping samples with Sendai expression from downstream expression meaningfully impacted overall findings.

Overall, our rigorous bioinformatic strategy adjusted for technical variation and batch effects, eliminated spurious samples, and flagged samples that were contaminated or had aberrant X-inactivation. This extensive analysis was motivated by the high level of intra-donor expression variation (see below), and eliminating these factors as possible explanations for this expression variation ultimately improved our ability to resolve SZ-relevant biology in our data set.

**COS RNA-Seq data cluster with existing data sets.** To assess the similarity of our hiPSC-NPCs and hiPSC-neurons to other hiPSC studies (by ourselves and others), as well as to post-mortem brain, we compared our data set to publicly available hiPSC, hiPSC-derived NPCs/neurons, and post-mortem brain homogenate expression data sets (Fig. 2). Hierarchical clustering indicated that similarity in expression profiles is largely determined by cell type (Fig. 2a). hiPSC-NPC and hiPSC-neuron data sets were more similar to prenatal samples than postnatal or adult post-mortem samples[20–22], which is consistent with previous reports[6,23–26]. hiPSC-NPCs and hiPSC-neurons, as well as post-mortem brain samples, cluster separately from hiPSCs, ESCs, fibroblasts and whole blood[12,20,27]. Despite being reprogrammed and differentiated through different methodologies, hiPSC-NPCs and hiPSC-neurons from the current study cluster with hiPSC-NPCs and hiPSC-neurons, respectively, generated previously in the same lab[2,28] and with hiPSC-NPCs and hiPSC-neurons from others[29], although some hiPSC-neurons[3] are more similar to prenatal brain samples from multiple brain regions[22]. Consistent with a differentiation paradigm from hiPSC to NPC to neuron, multidimensional scaling analysis (Fig. 2b) indicated that hiPSC-NPCs more resemble hiPSCs/hESCs than do hiPSC-neurons.

Genome-wide, hiPSC-NPCs and hiPSC-neurons express a common set of genes, so that expression differences between these cell types appear as changes in expression magnitude rather than activation of entirely different transcriptional modules (Supplementary Fig. 6). Yet this observation is also consistent with continuous variation in CTC, whereby the transcriptional signature of each cell type is present in each population at varying levels. Moreover, for both hiPSC-NPCs and hiPSC-neurons, genes that show high variance across donors in each cell type are enriched for brain eQTLs (Supplementary Fig. 7). Taken together, these two insights justified case-control comparisons within and between both hiPSC-NPCs and hiPSC-neurons.

**Large heterogeneity in cell type composition.** Given the substantial variability we observed between hiPSC-NPCs and hiPSC-neurons, even from the same individual (Supplementary Fig. 8), it seemed likely that inter-hiPSC and inter-NPC differences in differentiation propensity led to unique neural compositions in each sample. hiPSC-NPCs show extensive cell-to-cell variation in the expression of forebrain and neural stem cell markers[6] and 6-week-old neurons are comprised of a heterogeneous mixture of predominantly excitatory neurons, but also inhibitory and rare dopaminergic neurons, as well as astrocytes[7]. We hypothesized that CTC could be inferred using existing single cell RNA-Seq data sets and would enable us to (partially) correct for variation in differentiation efficiencies and account for some of the intra-individual expression variation.

**Fig. 1** COS hiPSC cohort reprogramming and differentiation. **a** Validated hiPSCs (from 14 individuals with childhood-onset-schizophrenia (COS) and 12 unrelated healthy controls) and NPCs (12 COS; 12 control individuals) yielded 94 RNA-Seq samples (11 COS; 11 control individuals). **b** Schematic illustration of the reprogramming and differentiation process, noting the yield at each stage. **c** Sex breakdown of the COS-control cohort. **d** Breakdown of SZ-associated copy number variants in the 11 COS patients with RNA-Seq data. **e** Representative qPCR validation of *NANOG, NESTIN*, and *SYN1* expression in hiPSCs (white bar), NPCs (light gray) and 6-week-old neurons (dark gray) from three individuals. **f** FACS analysis for pluripotency markers TRA-1-60 (left) and SSEA4 (right) in representative control (blue, $n = 17$) and COS (red, $n = 16$) hiPSCs. **g** FACS analysis for NPC markers SOX2 (left) and NESTIN (right) in control (blue, $n = 34$) and COS (red, $n = 37$) NPCs. **h** Representative images of NPCs (left) and 6-week-old forebrain neurons (right) from control (top) and COS (bottom). NPCs stained with SOX2 (red) and NESTIN (green); neurons stained with MAP2 (red). DAPI-stained nuclei (blue). Scale bar=50 μm. **i** Computational workflow showing quality control, integration with external data sets, computational deconvolution with Cibersort, decomposition multiple sources of expression variation with variancePartition, coexpression analysis with WGCNA, differential expression and concordance analysis

**Table 1 Number of individuals and cell lines at each step of experimental workflow**

| Experimental workflow | Total individuals | | Total hiPSC lines | | Total NPC lines | | Total neurons | |
|---|---|---|---|---|---|---|---|---|
| | control | COS | control | COS | control | COS | control | COS |
| Fibroblasts | 12 | 16 | – | – | – | – | – | – |
| hiPSCs | 12 | 14 | 35 | 40 | – | – | – | – |
| NPCs | 12 | 12 | 35 | 32 | 35 | 32 | – | – |
| RNA submitted | 11 | 11 | 20 | 22 | 24 | 23 | 24 | 23 |
| RNA-Seq QC passed | 9 | 10 | 17 | 18 | 20 | 18 | 20 | 18 |

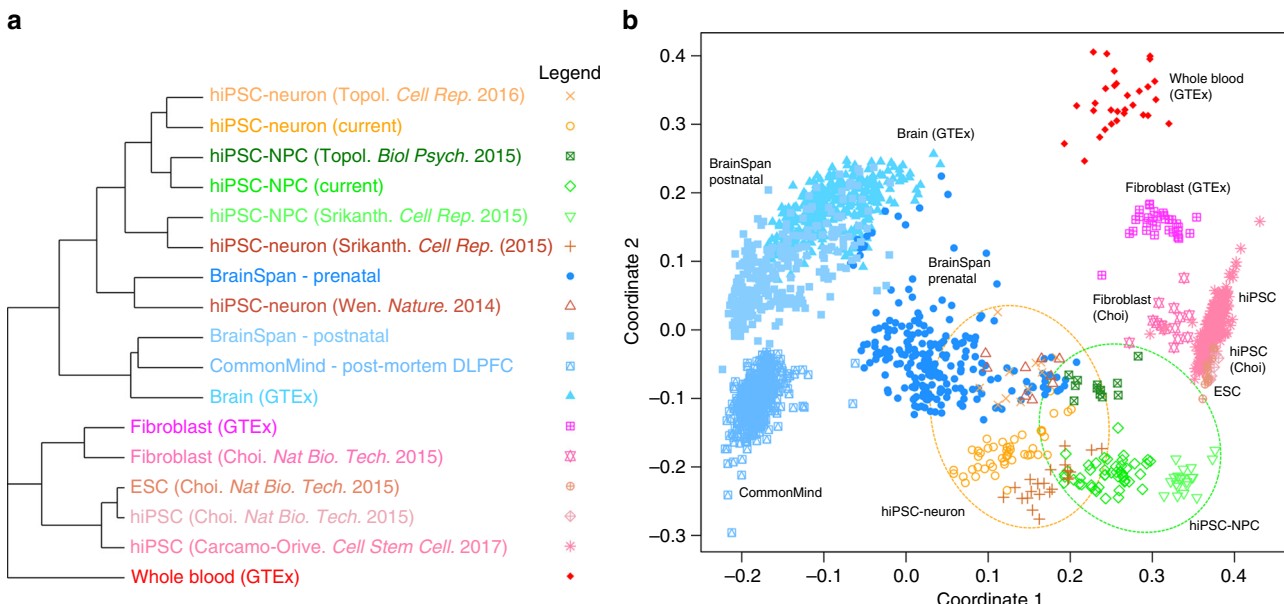

**Fig. 2** Cell type specificity of gene expression. **a** Summary of hierarchical clustering of 2082 RNA-Seq samples shows clustering by cell type. A pairwise distance matrix was computed for all samples, and the median distance between all samples in each category were used to create a summary distance matrix using to perform the final clustering. **b** Multidimensional scaling with samples colored as in **a**. hiPSC-NPCs from multiple studies are indicated in the green circle, and hiPSC-neurons from multiple studies are indicated in the orange circle

Bulk RNA-Seq analysis reflects multiple constituent cell types; therefore, we performed computational deconvolution analysis using CIBERSORT[30] to estimate CTC scores for each hiPSC-NPC and hiPSC-neuron sample (Fig. 3). A reference panel of single-cell sequencing data from mouse brain[31], mouse cell culture of single neural cells[32] and bulk RNA-Seq from hiPSC[27] was applied.

Overlaying CTC scores on a principal component analysis (PCA) of the expression data indicates that hiPSC-NPCs and hiPSC-neurons separate along the first principal component (PC), explaining 25.8% of the variance, and that the cell types have distinct CTC scores (Fig. 3a–c). As expected, hiPSC-neuron samples had a higher neuron CTC score than hiPSC-NPCs (mean increase = 0.06, $p < 1.05e\text{-}6$ by linear model) (Fig. 3a), while hiPSC-NPCs had a higher hiPSC CTC score (mean increase = 0.20, $p < 1.49e\text{-}31$ by linear model), consistent with a "stemness" signal (a neural stem cell profile was lacking from our reference) (Fig. 3b). Unexpectedly, hiPSC-neurons had a higher fibroblast$_1$ score (mean increase = 0.09, $p < 1.1e\text{-}6$ by linear model) (Fig. 3c). Rather than imply that there are functional fibroblasts within the hiPSC-NPC populations, we instead posit that this fibroblast signature is instead marking a subset of unpatterned, potentially non-neuronal cells[32]. Analysis of external NPC and neuron data sets indicates that these observations were reproducible, although there is substantial variability in CTC scores across data sets (Supplementary Fig. 9). Correction for CTC improved our ability to distinguish hiPSC-NPC and hiPSC-neuron populations; nonetheless, there remained substantial variability within both the hiPSC-NPCs and hiPSC-neurons that corresponded to CTC (Fig. 3d).

Not only is there significant overlap between fibroblast, mesenchymal and neural crest gene expression signatures (reviewed[33]), but both skin fibroblast preparations[34] and hiPSC-derived NPCs[35–37] show evidence of mesenchymal and/or neural crest contaminants. Therefore, it is important to consider the fibroblast$_1$ and fibroblast$_2$ signatures only as a tool with which to assess the variability in differentiation quality; high values for the "fibroblast signature" may well imply the presence of non-fibroblast contaminant(s) such as neural crest and/or mesenchymal cells. Supplementary Fig. 10 plots the expression of key neural crest[38,39] and mesenchymal[40] genes in our hiPSC-NPC and hiPSC-neuron data sets, as well as the reference panels.

The effect of CTC heterogeneity, likely due to the variation in differentiation efficiency, can be reduced by including multiple CTC scores in a regression model and computing the residuals. Using an unbiased strategy, we systematically evaluated which CTC score(s), when included in our model, most explained the variance in our samples. PCA on the residuals from a model including fibroblast$_1$ and fibroblast$_2$ CTC scores showed a markedly greater distinction between cell types, such that the first PC now explained 45.3% of the variance (Fig. 3e). Moreover, accounting for the CTC scores increased the similarity between

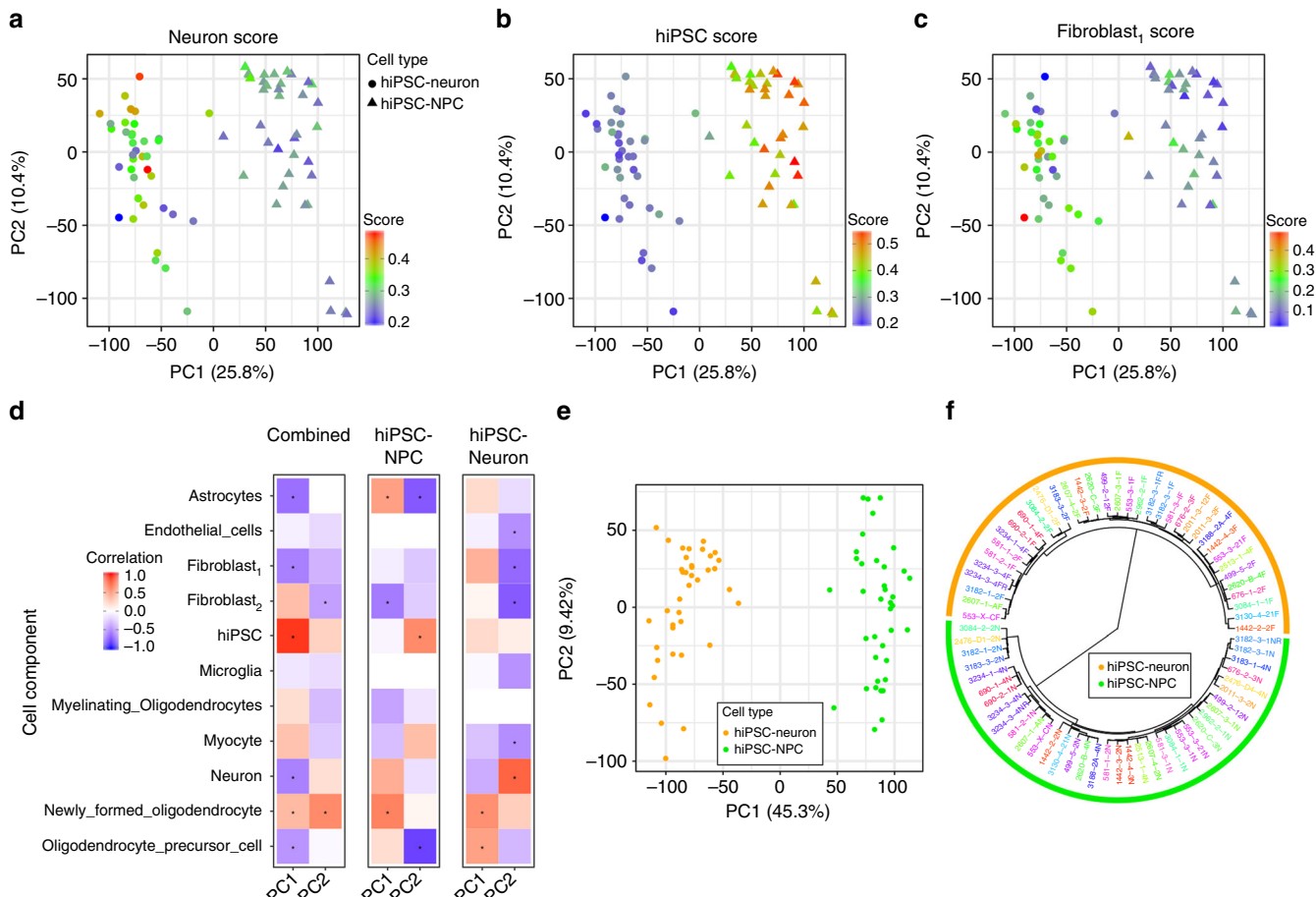

**Fig. 3** Variation in cell type composition contributes to gene expression variation. **a–c** Principal components analysis of gene expression data from hiPSC-NPCs (triangles) and hiPSC-neurons (circles) where samples are colored according to their cell type composition scores from cibersort for **a** neuron, **b** hiPSC, and **c** fibroblast$_1$ components. Color gradient is shown on the bottom right of each panel. **d** Correlation between 11 cell type composition scores for the first two principal components of gene expression data from all samples, only hiPSC-NPCs, and only hiPSC-neurons. Red indicates a strong positive correlation with a principal component and blue indicates a strong negative correlation. Asterisks indicate correlations that are significantly different from zero with a *p*-value that passes the Bonferroni cutoff of 5% for 66 tests. **e** Principal components analysis of expression residuals after correcting for the two fibroblast cell type composition scores. **f** Hierarchical clustering of samples based on expression residuals after correcting for the two fibroblast cell type composition scores

the multiple biological replicates generated from the same donor and resulted in less intra-individual variation within each cell type (Fig. 3f, Supplementary Fig. 11). Finally, accounting for CTC was necessary in order to see concordance with one of the adult post-mortem cohorts (see below).

**Characterizing known sources of expression variation**. As discussed above, gene expression (in our data set and others) is impacted by a number of biological and technical factors. By properly attributing multiple sources of expression variation, it is possible to (partially) correct for some variables. To decompose gene expression into the percentage attributable to multiple biological and technical sources of variation, we applied variancePartition[41] (Fig. 4). For each gene we calculated the percentage of expression variation attributable to cell type, donor, diagnosis, sex, as well as CTC scores for both fibroblast sets. All remaining expression variation not attributable to these factors was termed residual variation. The influence of each factor varies widely across genes; while expression variation in some genes is attributable to cell type, other genes are affected by multiple factors (Fig. 4a). Overall, and consistent with the separation of hiPSC-NPCs and hiPSC-neurons by the first PC, cell type has the

largest genome-wide effect and explained a median of 13.3% of the observed expression variation (Fig. 4b). Expression variation due to diagnosis (i.e., between SZ and controls) had a detectable effect in a small number of genes. Meanwhile, variation across the sexes was small genome-wide, but it explained a large percentage of expression variation for genes on chrX and chrY. Technical variables such as hiPSC technician, hiPSC date, NPC generation batch, NPC technician, sample name, NPC thaw and RIN explained little expression variation (Supplementary Fig. 12), especially compared to technical effects observed in previous studies[12,41].

Variation attributable to cell type heterogeneity across the CTC scores had a larger median effect than the variation across the 22 donors (fibroblast$_1$: 3.3%, fibroblast$_2$: 3.2%). The median observed variation across donor is 2.2%, substantially lower than reported in other data sets from hiPSCs[12,42] and other cell types[41]. By considering CTC in our model, the percentage of variation explained by donor significantly increased (median increase to 2.4%, *p* < 5.8e-62, one-sided paired Wilcoxon), indicating that cell type heterogeneity is an important source of intra-donor expression variation that obscures some inter-donor variation (i.e., case/control differences) of particular biological interest. Critically, there is no apparent diagnosis dependent variation in

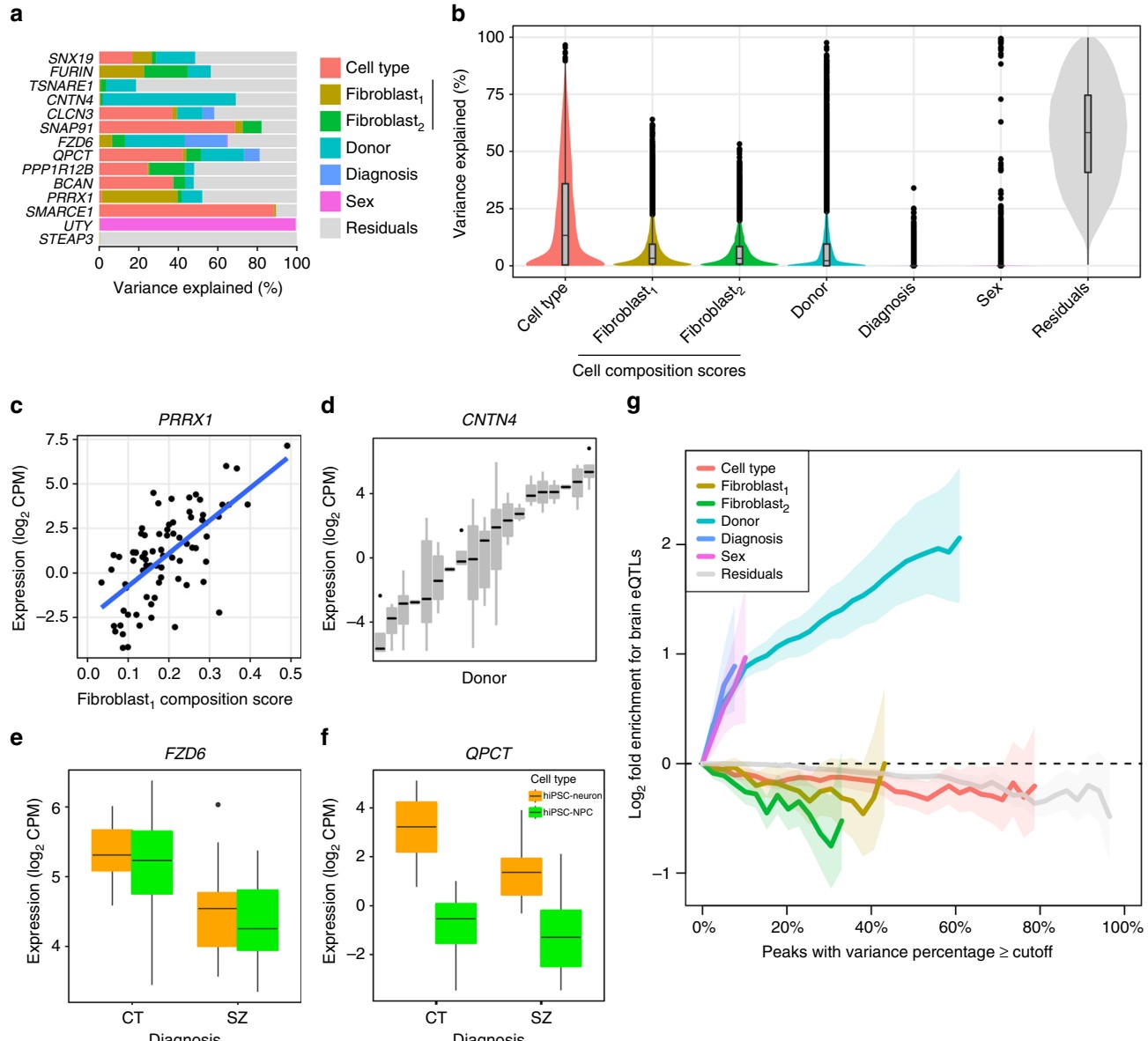

**Fig. 4** Decomposing expression variation into multiple sources. **a** Expression variance is partitioned into fractions attributable to each experimental variable. Genes shown include genes of known biological relevance to schizophrenia and genes for which one of the variables explains a large fraction of total variance. **b** Violin plots of the percentage of variance explained by each variable over all the genes. **c–f** Expression of representative genes stratified by a variable that explains a substantial fraction of the expression variation. **c** *PRRX1* plotted as a function of the fibroblast₁ cell type composition score. **d** *CNTN4* stratified by donor. **e** *FZD6* stratified by disease status and cell type. **f** *QPCT* stratified by disease status and cell type. **g** Genes that vary most across donors are enriched for brain cis-eQTLs. Fold enrichment (log₂) for the 2000 top cis-eQTLs discovered in post mortem dorsolateral prefrontal cortex data generated by the CommonMind Consortium[21] shown for six sources of variation, plus residuals. Each line indicates the fold enrichment for genes with the fraction of variance explained exceeding the cutoff indicated on the *x*-axis. Shaded regions indicate the 90% confidence interval based on 10,000 permutations of the variance fractions. Enrichments are shown on the *x*-axis until less that 100 genes pass the cutoff

CTC (Supplementary Fig. 13). By compensating for CTC, we prevent variation in neuronal differentiation between hiPSCs from overriding some of the donor-specific gene expression signature that is the central focus of patient-derived cell culture models.

The percentage of expression variation explained by each factor has a specific biological interpretation. *PRRX1* is known to function in fibroblasts[43,44] and variation in the fibroblast₁ CTC score explains 38.3% of expression variant in this gene (Fig. 4c). Expression of *CNTN4* is driven by an eQTL in brain tissue that corresponds a risk locus for SZ[21]. In our data, *CNTN4* has 67.4% expression variation across donors suggesting that this variation

is driven by genetics (Fig. 4d). Genes that vary across diagnosis correspond to differentially expressed genes, including *FZD6*, a WNT signaling gene linked to depression[45], (Fig. 4e) and *QPCT*, a pituitary glutaminyl-peptide cyclotransferase that has been previously associated with SZ[46] (Fig. 4f).

Genes that vary across donors were enriched for eQTLs detected in post-mortem brain tissue[21] (Fig. 4g), meaning that observed inter-individual expression variation reflected genetic regulation of expression. Conversely, genes with expression variation attributable to cell type (CTC scores) are either neutral or depleted for genes under genetic control, indicating that variation in CTC was either stochastic or epigenetic, but did not

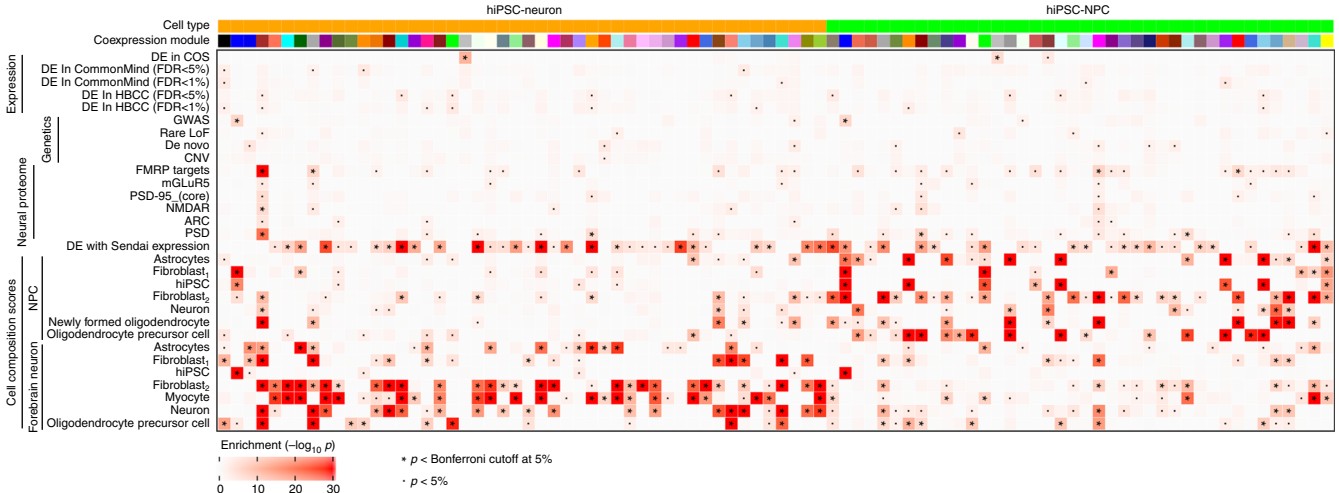

**Fig. 5** Clustering of genes into coexpression modules reveals module-specific enrichments. Enrichment significance ($-\log_{10}$ p-values from hypergeometric test) are shown for coexpression modules from hiPSC-NPCs and hiPSC-neurons. Each module is assigned a color and only modules with an enrichment passing the Bonferroni cutoff in at least one category is shown. Enrichments are shown for gene sets from RNA-Seq studies of differential expression between schizophrenia and controls; genetic studies of schizophrenia, neuronal proteome[21]; and cell composition scores from hiPSC-NPCs and hiPSC-neurons in this study. p-values passing the 5% Bonferroni cutoff are indicated by '\*', and p-values < 0.05 are indicated with '.'

reflect genetic differences between individuals. Finally, the high percentage of residual variation not explained by factors considered here suggests that there are other uncharacterized sources of expression variation, including stochastic canalization effects or unexplained variation in CTC.

**WGCNA analysis identifies modules enriched for SZ and CTC.** Genes with similar functions are known to share regulatory mechanisms and so are often coexpressed[47]. We used weighted gene coexpression network analysis (WGCNA)[48] to identify modules of genes with shared expression patterns (Fig. 5, Supplementary Data 7). Genes were clustered into modules of a minimum of 20 genes, and each module was labeled with a color (Supplementary Fig. 14). Genes that did not form strong clusters were assigned to the gray module. Analysis was performed separately in hiPSC-NPCs and hiPSC-neurons; each module was evaluated for enrichment of genes for multiple biological processes. Many modules were highly enriched for genes that were significantly correlated with CTC scores at FDR < 5%, underscoring the genome-wide effects of cell type heterogeneity. Genes that were differentially expressed between cases and controls in this study (see below) were enriched in the gray modules in both hiPSC-NPCs (OR = 1.99, p < 1.45e-5) and hiPSC-neurons (OR = 3.44, p < 5.04e-12, hypergeometric test), indicating that in this data set, differentially expressed genes did not form a coherent structure but are instead widely distributed. Genes identified by genetic studies (i.e., common variants, CNVs, rare loss of function and de novo variants) and case/control signatures from two post-mortem data sets (the CommonMind Consortium (CMC)[21] and the NIMH Human Brain Collection core (HBCC)) showed moderate enrichment in many modules, but did not strongly overlap with the gray modules enriched for differentially expressed genes from this study. Finally, gene sets corresponding to the neural proteome show the strongest enrichment in the brown module from hiPSC-neurons, including, the targets of FMRP (OR = 4.06, p < 2.84e-40) and genes involved in postsynaptic density (OR = 3.35, p < 5.45e-22).

**Differential expression between COS and control hiPSC-NPCs and hiPSC-neurons.** The central objective of this study was to determine if a gene expression signature of SZ could be detected in an experimentally tractable cell culture model (Fig. 6). Due to the "repeated measures" study design where individuals are represented by multiple independent hiPSC-NPC and hiPSC-neuron lines, we used a linear mixed model by applying the duplicateCorrelation function in our limma/voom analysis[49]. This approach is widely used to control the false positive rate in studies of repeated measures and its importance in hiPSC data sets was recently emphasized[15].

Differential expression analysis between cases and controls in hiPSC-NPCs (Fig. 6a) identified 1 gene with FDR < 10% and 5 genes with FDR < 30%; analysis in hiPSC-neurons (Fig. 6b) identified 1 gene with FDR < 10% and 5 genes with FDR < 30% (Supplementary Data 8).

While plausible candidates such as *FZD6* and *QPCT* were differentially expressed, gene set enrichment testing did not implicate a coherent set of pathways (Supplementary Data 9). As SZ is a highly polygenic disease and this data set is underpowered due to the small sample size[21], we expected the disease signal to be subtle and distributed across many genes. Despite performing extensive analysis using sophisticated statistical methods built on top of the limma/voom framework[50] that incorporated genes that were not genome-wide significant and using permutations to empirically set the significance cutoff (see Methods), we failed to identify a coherent biological enrichment. Nonetheless, there was an unexpected concordance in the differential expression analysis between COS and control hiPSC-NPCs and hiPSC-neurons, which showed remarkably similar $\log_2$ fold changes (Fig. 6c). Moreover, no genes had $\log_2$ fold changes that were statistically different in the two cell types, although we were underpowered to detect such differences.

Overall, our differential expression analysis demonstrated that case-control hiPSC-based cohorts remain under-powered to resolve biologically coherent SZ-associated processes. Nonetheless, the concordance in the disease signature identified in hiPSC-NPCs and hiPSC-neurons implies that future studies could focus on just one cell type.

**Concordant differential gene expression with post-mortem data sets.** While it is well-understood that all hiPSC-based studies of SZ remain under-powered due to small sample sizes and

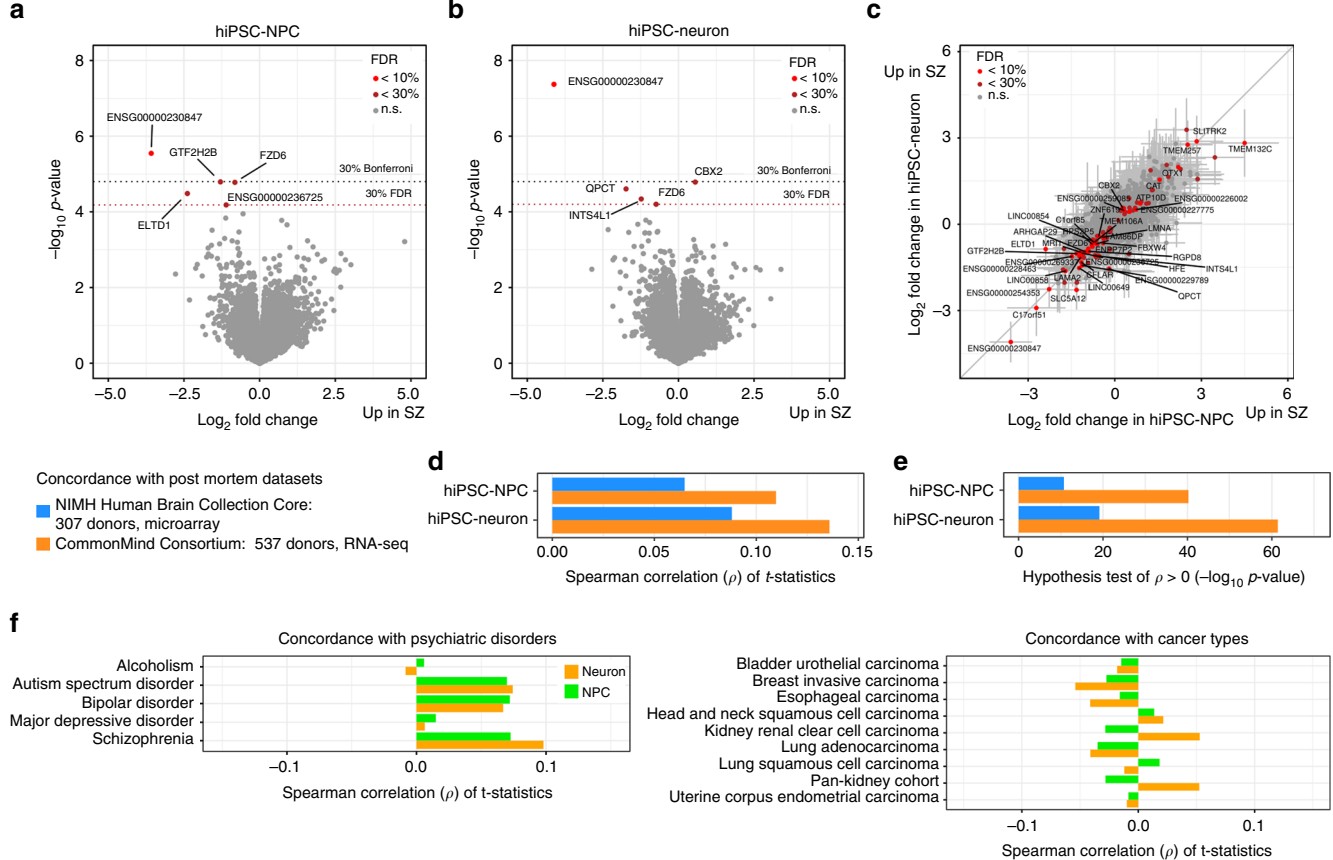

**Fig. 6** Differential expression between schizophrenia and controls. **a**, **b** Volcano plot showing log$_2$ fold change between cases and controls and the −log$_{10}$ p-value for each gene in **a** hiPSC-NPC and **b** hiPSC-neuron samples. Genes are colored based on false discovery rate: light red (FDR < 10%), dark red (FDR < 30%), gray (n.s. not significant). Names are shown for genes with FDR 30%. Dotted gray line indicates Bonferroni cutoff corresponding to a p-value of 0.30. Dashed dark red line indicates FDR cutoff of 30% computed by qvalue (Storey[82]). **c** Log$_2$ fold change between cases and control in hiPSC-NPCs (x-axis) compared to log$_2$ fold change between cases and controls in hiPSC- neurons (y-axis). Genes are colored according to differential expression results from combined analysis of both cell types: light red (FDR < 10%), dark red (FDR < 30%), gray (n.s. not significant). Error bars represent 1 standard deviation around the log$_2$ fold change estimates. **d**, **e** Analysis of concordance between differential expression results of schizophrenia vs. controls from the current study and two adult post mortem cohorts[21]. Concordance is evaluated based on spearman correlation between t-statistics from two data sets. **d** Spearman correlation between t-statistics from the current study (from hiPSC-NPCs and -neurons) and the two post mortem cohorts. **e** −log$_{10}$ p-values from a one-sided hypothesis test for the Spearman correlation coefficients from **d** being greater than zero. **f** Concordance of t-statistics with differential expression results from case-control analysis of five psychiatric diseases[53] and tumor-normal analysis of nine cancer types[54]

polygenic disease architecture, what is less appreciated is that post-mortem approaches are similarly constrained. Using allele frequencies from the Psychiatric Genetics Consortium data, the median number of subjects needed to obtain 80% power to resolve genome-wide expression differences in SZ cases was estimated to be ~28,500, well beyond any existing data set[21]. Nonetheless, we evaluated the concordance of our data set with the findings of two much larger post-mortem studies, CMC: RNA-Seq from 537 donors; NIMH HBCC, microarrays from 307 donors) by computing the correlation in t-statistics from the differential expression analysis between cases and controls.

The Spearman correlation between our hiPSC-NPC results and the CMC and NIMH HBCC results were 0.108 and 0.0661, respectively; for the hiPSC-neurons results, the correlations were 0.134 and 0.0896, respectively (Fig. 6d, Supplementary Figs. 15 and 16). These correlations were highly statistically significant (Fig. 6e) for both hiPSC-NPCs: p < 4.6e−40 and 7.8e−12 for CMC and HBCC, respectively; and for hiPSC-neurons: p < 6.7e−61 and 1.6e−20 respectively (Spearman correlation test). Similar results were obtained by using Pearson correlation and by evaluating the concordance using the log$_2$ fold changes from each data set (Supplementary Figs. 15 and 16). This stronger concordance of

hiPSC-neurons (relative to hiPSC-NPCs) with post-mortem findings is consistent with the hypothesis that neurons are the cell type most relevant to SZ risk[51], but our ability to resolve it is perhaps surprising in that neurons are estimated to comprise a minority of the cells in brain homogenate[52]. To a lesser extent, this concordance was also detected in ASD and BD post-mortem data sets, but not in other neuropsychiatric disorders[53] such as alcoholism and major depression disorder, or a variety of cancer types[54] (Fig. 6f), indicating the specificity of our results.

While the concordance with CMC was observed when correcting for any set of CTC scores (or none), the concordance with HBCC was only apparent when correcting for the fibroblast$_1$ CTC score (Supplementary Fig. 17). This illustrates the importance of accounting for CTC and the fact that concordance can be obscured by biological sources of expression variation. The genes for which the differential expression signal was boosted by accounting for the fibroblast$_1$ score were enriched for brain and synaptic genesets, including specific biological functions such as FMRP and mGluR5 targets (Supplementary Figs. 18 and 19).

Given the degree of concordance in the SZ differentially expressed genes between the hiPSC-NPCs, hiPSC-neurons, CMC and NIMH HBCC data sets (Fig. 6d, e), the lack of enrichment of

the CMC or NIMH HBCC differentially expressed genes in the "gray module" of our coexpression analysis (Fig. 5) is noteworthy. Although the concordance and coherence of the signal between hiPSC-NPCs and hiPSC-neurons with two post-mortem data sets was relatively low, we believe this reflects the small sample size and low power of our current study and predict that both will increase with expanding sample sizes in future studies.

## Discussion

SZ is a complex genetic disease arising through a combination of rare and common variants. Recent large-scale genotyping studies have begun to reveal the extent to which SZ risk reflects rare copy number variants (CNVs)[11] and coding mutations[55], as well as common single nucleotide polymorphisms (SNPs) with small effect sizes[56]. The strongest finding to date from these genetic studies is that SZ-associated variants are enriched for pathways primarily associated with synaptic biology[55,57]. Although > 50 post-mortem gene expression studies of SZ have been reported, the results have been inconsistent, likely owing to the small sample sizes involved[21]. The largest of these, comparing brain tissue from 258 subjects with SZ and 279 controls did not find evidence for case–control differential expression among the implicated SZ risk genes; moreover, by modeling both the allele frequencies and the predicted allelic effects on gene expression, they predicted the median number of subjects needed to obtain genome-wide power (80%) to be ~28,500[21]. This issue of small sample sizes is not unique to post-mortem studies, and may be

exacerbated in hiPSC-based experiments through the variability that arises as a result of the reprogramming and differentiation processes. We established an hiPSC cohort of COS patients[58–62], testing our ability to model gene expression changes associated with both common and rare variants in vitro. While other studies have focused on SZ cohorts comprised of relatively few individuals with rare mutations[3–5], we sought to determine to what extent a larger cohort captured the expression signature of polygenic SZ, focusing on COS due to the higher genetic burden of both rare and common variants in these patients.

The goal of studying patient-derived cell culture models is to develop an experimentally tractable platform that recapitulates a donor-specific gene expression signature. Retaining this donor-specific signature is essential to studying case-control differences. In two recent studies of hiPSCs, variance across donors explained a median of ~6[55] and 48.8%[12] of expression variation, while the effect of donor was much smaller (2.2%) in this study. We hypothesize that donor effects are reduced due to stochastic noise in the differentiation from hiPSCs to neurons; it remains to be established whether different hiPSC-derived cell types will retain more or less donor signal over the course of differentiation. In our data set, while genes with high expression variation across donors were enriched for eQTLs detected in post-mortem brain, substantial expression variation within donors obscured some biological signal. In order to identify biological or technical variations that explained this intra-donor expression variation, we implemented a quality control pipeline to detect sample mislabeling, cell culture contamination, residual Sendai virus

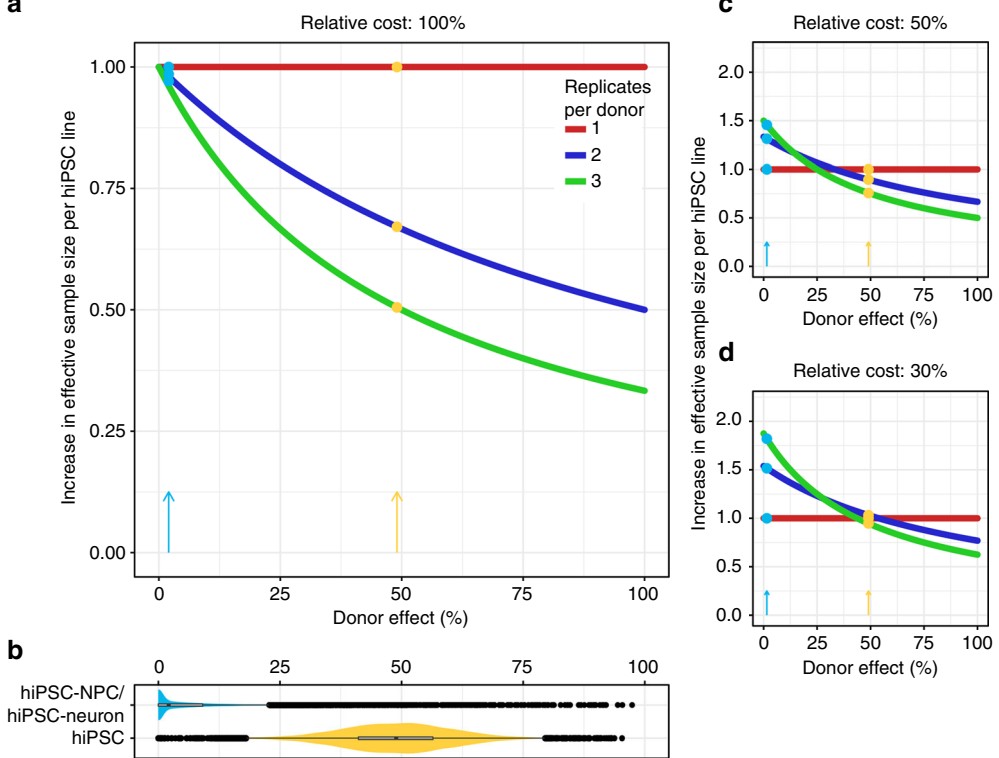

**Fig. 7** Maximizing power in hiPSC studies depends on relative costs and the fraction of expression variation across donors. **a** The increase in effective sample size (ESS) for each additional hiPSC line added to the data set shows as a function of the donor effect when the cost or an additional hiPSC line is the same as the cost for an additional donor. The increase in ESS is constant for the first replicate from a donor, while the contribution of the second or third replicates depend heavily on the donor effect. Colored points and arrows indicate the increase in ESS based on the donor effect from the current study (blue) and hiPSCs (orange)[12]. **b** Violin plots show the full distribution and median donor effect computed by variancePartition for the current study (blue) and hiPSCs (orange). The median values across all genes correspond to the colored arrows and points in the other panels. **c** Plot of ESS as in **a** but where the relative cost of an additional hiPSC line is 50% of the cost of an additional donor. **d** Plot of ESS as in **a** but where the relative cost of an additional hiPSC line is 30% of the cost of an additional donor

expression, incomplete X-inactivation and batch effects in sample processing; however, it was only accounting for variation in CTC that significantly decreased intra-donor variation.

The persistent expression of exogenous reprogramming factors, particularly *c-MYC*, despite the use of sendai viral non-integrative methods has been previously reported[63,64] and may reflect the variation of vector replication between cell lines, as well as a potential growth advantage of c-MYC expressing cells[64]. Although standard non-integrative methodologies rely upon passive and inefficient omission for the loss of sendai viral vectors[64], new methods, such as auto-erasable Sendai virus vectors[65], should facilitate the generation of truly transgene-free hiPSCs.

Given the challenges of low statistical power, substantial intra-donor variation, and the range of complicating factors that can obscure the disease signal, future hiPSC-based studies of human disease should be carefully designed to maximize power. One particular challenge affecting many studies is the tradeoff between increasing the number of biological replicates and increasing the number of donors. The statistical concept of "effective sample size" (ESS) addresses this issue directly and indicates that the tradeoff is dependent on the cost per donor and per hiPSC line in addition to the fraction of expression variation explained by donor (Supplementary Note 1). When a study includes multiple correlated samples from the same donor, the ESS is defined as the sample size of a study with equivalent power composed of only independent samples (Fig. 7). When the cost for each donor and each additional replicate are equal, adding an additional donor will increase the ESS by one unit (Fig. 7a), while adding an additional sample from an existing donor will increase the ESS by only a fraction of a unit because a sample correlated with it is already in the data set. The contribution of each additional sample is determined by the donor effect. Therefore, when biological replicates from the same donor are very correlated, the increase in ESS can be small. Conversely, adding replicates when there is high intra-donor variability (i.e., a low donor effect) can have a larger increase on ESS. The fact that the donor effect in the current study is lower than in previous hiPSC studies[12,42] affects the contribution of each additional sample to the ESS (Fig. 7b). When the costs for an additional hiPSC line are less than the cost of an additional donor, the calculus changes in favor of including additional biological replicates (Fig. 7c, d). We have developed a public website (http://gabrielhoffman.shinyapps.io/design_ips_study/) that computes the ESS in order to design a study to maximize power. These calculations consider constraints on either total budget or number of donors, as the relative cost and donor effect change. Overall, our conclusion is that the best way to maximize ESS, while controlling the false positive rate, is often to use one hiPSC line per donor and increase the number of donors, rather than using multiple replicate clones from a smaller set of donors[15,66,67].

In addition to maximizing cohort ESS, future studies will benefit from decreasing intra-donor expression variation by optimizing neuronal differentiation/induction protocols to focus on decreasing cellular heterogeneity (rather than increasing total yield). The generation of single-cell sequencing data sets from hiPSC-NPCs and/or hiPSC-neurons will further yield a custom reference panel with which to improve CTC deconvolution. In fact, our results suggest that to maximize ESS while minimizing associated costs, it may be sufficient to focus on a single cell type, hiPSC-neurons rather than hiPSC-NPCs. Given our improved understanding of the inherent challenges associated with studying highly polygenic diseases as well as the biological constraints encountered with hiPSC-based models here, disease signal will be further improved by reducing disease heterogeneity through focusing on cohorts of patients with shared genetic variants and/ or the genetic engineering of isogenic hiPSC lines to introduce or repair SZ-relevant variants.

Despite our relatively small sample size, we were able to identify a subtle but statistically significant concordance between both COS hiPSC-NPCs and hiPSC-neurons with two recent SZ post-mortem cohorts[21], an effect that was strongest in hiPSC-neurons. Yet this shared biology did not yield enrichments at the pathway or network level in the diagnosis-dependent differentially expressed genes observed between hiPSC-NPCs and hiPSC-neurons with either post-mortem data set. Moving forward, increasing the sample size of hiPSC-based cohorts may improve this concordance and biological coherence. Alternatively, it is possible that many SZ-associated processes are not present in simple monolayer hiPSC-NPC and hiPSC-neuron populations; relevant aspects of SZ biology may only be detected through activity-dependent processes arising from complex neuronal circuitry, following oligodendrocyte myelination, astrocyte support or microglia pruning, or after exposure to neuroinflammation or environmental stimuli. While the best strategy to improve biological significance is to strive to enhance the complexity of hiPSC-based models, the surest approach to improve the power of case-control comparisons is to integrate a growing number of post-mortem and hiPSC studies. In either case, to facilitate improved sharing between stem cell laboratories, all hiPSCs have already been deposited at a repository. We urge widespread sharing of all RNA-Seq data and reproducible scripts, and so make ours available.

## Methods

**hiPSC derivation and differentiation.** Description of COS cohort: Childhood-onset-schizophrenia (COS) is reliably diagnosed in children using unmodified DSM criteria[68]; there is no clinical, neuroimaging, pharmacological or genetic evidence to suggest that COS is a distinct disorder (reviewed[69–71]). This is an unusually well-characterized cohort, with medication-free in-patient observation used for diagnosis[72]. The following clinical information was collected: gender, age at biopsy, developmental history, age of symptom onset, IQ, number of hospitalizations as a measure of disease severity, positive and negative symptom scale, diagnostic screening by Comprehensive Assessment of Symptoms and History (CASH), attention tests, current antipsychotic treatment, clozapine responsiveness, and substance abuse history.

Patients with COS, unaffected family members, and unrelated controls were recruited into a longitudinal study by Dr Judith Rapoport at the NIMH[9,10]; many had skin biopsies completed. The Rapoport laboratory generously provided fibroblasts, from which 14 cases and 12 controls were reprogrammed. COS cases: NSB499 (male), NSB581 (male), NSB676 (female), NSB1251 (male), NSB1275 (female), NSB1358 (male), NSB1442 (male), NSB1804 (female), NSB2011 (female), NSB2476 (female), NSB2484 (female), NSB2513 (male), NSB2620 (male), NSB2962 (male). Controls: NSB553 (male), NSB690 (male), NSB2607 (male), NSB3084 (male), NSB3113 (female), NSB3121 (female), NSB3130 (male), NSB3158 (female), NSB3182 (female), NSB3183 (female), NSB3188 (female), NSB3234 (male). All fibroblast samples had IlluminaOmni 2.5 bead chip genotyping[9,10], PsychChip and exome sequencing completed. The PsychChip genotyping data were used to calculate the polygenic risk score for each individual in this study; polygenic risk scores and SZ-relevant CNVs are listed in Supplementary Data 1 and 2. All adult subjects (and parents of minor subjects) provided written and informed consent for skin biopsies, hiPSC reprogramming and genetic analyses. Minor subjects provided written and informed assent. All work was reviewed by the Internal Review Board of the Icahn School of Medicine at Mount Sinai. This work was also reviewed by the Embryonic Stem Cell Research Oversight Committee at the Icahn School of Medicine at Mount Sinai.

HFs were cultured on 0.1% (w/v) gelatin coated plates in HF medium (DMEM (ThermoFisher Scientific), 20% (v/v) FBS (Corning)). Replicating 90% confluent HFs were reprogrammed using a modified protocol; briefly, HFs were transduced with Cytotune® Sendai viruses (ThermoFisher Scientific) expressing *OCT4*, *SOX2*, *KLF4* and *c-MYC*, as per lot specifications. Cells from a single well of a six-well plate were split 1:2, 1:3 and 1:10 onto 10-cm plates containing $1\times10^6$ mouse embryonic fibroblasts (mEFs; GlobalStem). Cells were switched to hiPSC media (DMEM/F12, 20% KO-Serum Replacement (v/v), 1% (v/v) GlutaMAX, 1% (v/v) nonessential amino acids (NEAA), 55 µM β-mercaptoethanol (β-me) (all ThermoFisher Scientific) and 20 ng ml$^{-1}$ FGF2 (R & D Systems, 233-FB-10)) and fed daily. hiPSC colonies were manually picked and clonally plated onto 24-well mEF-coated plates. hiPSC lines were maintained on mEFs in hiPSC media; at early passages, hiPSCs were split using manual passaging and at higher passages, hiPSC were enzymatically passaged with Collagenase (1 mg/ml in DMEM) (Sigma) until

cryopreservation in cold freezing media (hiPSC media containing 10% DMSO). Individual hiPSC lines were validated using TRA-1-60 and SSEA-4 flow cytometry and NANOG, TRA-1-60, SOX2 and OCT4 immunocytochemistry. G-banded karyotyping was performed by WiCell Cytogenetic Services. The differentiation potential of the hiPSCs derived in this study was confirmed by RT-PCR for markers of the three germ layers following spontaneous differentiation of a subset of the lines into embryoid bodies. Routine (every 2–4 weeks) mycoplasma testing was conducted using the MycoAlert Mycoplasma detection kit (Lonza); all cells used in this study and were found to be negative.

NPCs were generated from unique hiPSC lines with had normal karyotypes, and then passed NPC quality control based on immunocytochemistry and FACS for SOX2 and NESTIN levels. NPCs were derived, as previously described[6] and maintained at high density, grown either growth factor reduced Matrigel (BD Biosciences) coated plates in NPC media (Dulbecco's Modified Eagle Medium/Ham's F12 Nutrient Mixture (ThermoFisher Scientific), 1x N2, 1x B27-RA (ThermoFisher Scientific) and 20 ng ml$^{-1}$ FGF2 and split 1:3 every week with Accutase (Millipore, Billerica, MA, USA). NPCs were dissociated with Accutase and plated at $2.0 \times 10^5$ cells per cm$^{-2}$ in NPC media onto growth factor reduced Matrigel-coated plates. For neuronal differentiation, media was changed to neural differentiation medium (DMEM/F12, 1xN2, 1xB27-RA, 20 ng ml$^{-1}$ BDNF (Peprotech), 20 ng ml$^{-1}$ GDNF (Peprotech), 1 mM dibutyryl-cyclic AMP (Sigma), 200 nM ascorbic acid (Sigma) and 1 µg ml$^{-1}$ laminin (ThermoFisher Scientific) 1–2 days later. NPC-derived neurons were differentiated for 6 weeks.

**FACS.** hiPSCs were labeled with TRA-1-60-488 (5 µl per $1 \times 10^6$ cells, BioLegend #330613) and SSEA4-647 (5 µl per $1 \times 10^6$ cells, BioLegend #33407) in 1% (w/v) BSA for 45 min at 4 °C before being washed with 1× PBS and resuspended in FACS buffer (1× PBS (no Mg$^{2+}$/Ca$^2$) containing 1% (v/v) BSA and TO-PRO*3 (1 µM, ThermoFisher Scientific) and filtered using a 40 µm filter (BD Biosciences). NPCs were dissociated using Accutase, fixed for 10 min in 4% paraformaldehyde (PFA), permeabilized and blocked with 0.5% (v/v) Triton (Sigma)/1% (w/v) bovine serum albumin (BSA, Sigma) in PBS and labeled with NESTIN-647 (20 µl per $1 \times 10^6$ cells, BD Biosciences #560393) and SOX2-488 (0.25 µg per $1 \times 10^6$ cells, BioLegend #656110) antibodies overnight at 4 °C before being washed with PBS and resuspended in FACS buffer (1× PBS (no Mg$^{2+}$/Ca$^2$) containing 1% (v/v) BSA and TO-PRO*3 (1 µM, ThermoFisher Scientific) and filtered using a 40 µm filter (BD Biosciences). Cytometry was performed using a LSR-II or FACS Canto (BD Biosciences) and analysis was performed using Flowjo (v8.7.3, Treestar).

**qPCR.** Total RNA was extracted using Trizol following the manufactures instructions. Transcript analysis was carried out using a QuantStudio™ 7 Flex Real-Time PCR System using the Power SYBR green RNA-to-Ct RT-qPCR kit for primers (all ThermoFisher Scientific). Around 50 ng of RNA template was added to the PCR mix. (ThermoFisher Scientific). qPCR conditions were as follows, 48 °C for 15 min, 95 °C for 10 min followed by 40 cycles (95 °C for 15 s, 60 °C for 60 s). Primers used as follows: *NANOG* (f: CAGTCTGGACACTGGCTGAA, r: CTCGCTGATTAGGCTCCAAC), *NESTIN* (f: GAAACAGCCATA-GAGGGCAAA, r: TGGTTTTCCAGAGTCTTCAGTGA), *SYN1* (f: GCA AGG ACG GAA GGG ATC ACA TCA, r: CCTGAGCCATCTTGTTGACCACGA), *ACTIN* (f: TGTCCCCCAACTTGAGATGT, r: TGTGCACTTTTATT-CAACTGGTC), *GAPDH* (f: AGGGCTGCTTTTAACTCTGGT, r: CCCCACTT-GATTTTGGAGGGA). Data were analyzed using GraphPad PRISM 6 software. Values are expressed as mean ± SEM.

**RNA sequencing.** RNA Sequencing libraries were prepared using the Kapa Total RNA library prep kit with ribo-depletion and strand specific cDNA library construction (Kappa Biosystems). Paired-end sequencing reads (125 bp) were generated on an Illumina HiSeq2000 platform (New York Genome Center).

**RNA-Seq processing.** RNA-Seq reads were aligned to GRCh37 with STAR v2.4.0g1[73]. Uniquely mapping reads overlapping genes were counted with featureCounts v1.4.4[74] using annotations from ENSEMBL v70. All analysis used log$_2$ counts per million (CPM) following TMM normalization[75] implemented in edgeR v3.14.0[76] unless stated otherwise. Genes with over > 1 counts per million in at least 30% of the experiments were retained.

**Identity checking.** Variant concordance analysis was performed to identify instances where samples labeled as being from the same donor are discordant based on variant calls. Variants were called from the RNA-Seq BAM files using GATK 3.4[77] following best practices to produce gVCF files. These files were merged using the GATK CombineVCFs functionality. The resulting VCF was then merged with variants from whole exome sequencing and PsychChip array from the same donors. Variant concordance between all pairs of samples was evaluated with bcftools v1.3. Discordant samples were relabeled when possible, otherwise they were excluded from downstream analysis.

**Contamination analysis.** VeryifyBamID[17] compares a BAM file from a sequencing experiment (i.e., RNA-Seq, or whole exome sequencing) to a set of reference genotypes to identify sample contamination. The software estimates the contamination percentage for each sample using a sophisticated statistical model. Each RNA-Seq BAM file was analyzed with verifyBamID using a VCF from either the PsychChip and whole exome sequencing data as the reference set. Results from both analyses were very similar. This method was originally designed for DNA sequencing where variants calls have a much lower error rate than from RNA-Seq. For this reason, multipotency data are expected to have increased contamination estimates even under the null model of no contamination.

**Analysis of gene expression within CNV regions.** CNV coordinates where stored in a BED file and genes overlapping these regions were identified in R. Gene expression residuals were computed by fitting log$_2$ CPM for each in a linear model in order to remove the effect of cell type. Z-scores for each gene were computed by subtracting the mean and dividing by the standard deviation for each gene. Expression outliers were identified based on extreme z-scores.

**Sendai virus detection and quantification.** Reads that did not map to the human genome with STAR were saved in a separate FASTQ file and Trinity[18] was used to perform de novo assembly of these reads. Trinity was run with flags—no_r-un_chrysalis—no_run_butterfly and otherwise with default settings were used. This produced a FASTA file of de novo contigs for each RNA-Seq experiment. Bowtie2 v2.1.0[78] was used to index this FASTA file and TopHat2 v2.0.6[79] was used to align reads from the RNA-Seq FASTQ to the de novo contigs. This step quantifies how many reads correspond to each contig. Next, each contig was aligned to a database of complete viral genes from NCBI using BLAST[80]. The results were filtered to retain only contigs aligning to the Sendai virus genome sequence (GenBank: AB855655.1). Note that the specific Sendai virus used in the iPSC reprogramming has been engineered to incorporate four human transcription factors, and the genome sequence is not available. Therefore we used AB855655.1 as a proxy. Finally, reads corresponding to contigs that align to the Sendai genome were counted for each RNA-Seq experiment and these values were included in downstream analysis.

**Cell composition analysis.** Cell type composition scores were computed using CIBERSORT v1.04[30] using default settings on the web interface. CIBERSORT uses a machine learning approach to estimate the cellular composition of each sample based on the expression profiles of a set of reference cell types. The reference set was constructed based on biological expectations of the constituent cell types.

- Single cell RNA-Seq from mouse brain[31]: Astrocytes, Neuron, Oligodendrocyte Precursor Cell, Newly Formed Oligodendrocyte, Myelinating Oligodendrocytes, Microglia, Endothelial Cells
- Single cell RNA-Seq from mouse cell culture from direct reprogramming from mouse embryonic fibroblast to neuron[32]. We included untreated mouse embryonic fibroblast (here termed fibroblast$_2$). Cells were transformed with Ascl1, Brn2 and Myt1l and cultured for 22 days. Single cells were sequenced, clustered computationally, and annotated based on characterizing gene expression patterns. Cells with annotated based on expression of know genes as either: Neuron, Myocyte, Fibroblast (here termed fibroblast$_1$).
- Bulk RNA-Seq from hiPSC[27].

Since single-cell expression data can be very noise, multiple examples of each cell type were included in the reference panel and the component scores were summed for each cell type. This analysis was performed multiple times with different representatives of each cell type included each time to ensure the results were robust.

For each sample, CIBERSORT reports the estimated percent composition for each cell type in the reference panel. However, the scale of these percentages is sensitive to the other cell types included in the panel and is often not biologically plausible. For example, while the hiPSC composition of NPCs is estimated to be ~35–57%, this not biologically realistic because (i) hiPSCs cannot survive in NPC culture conditions, (ii) hiPSCs replicate more quickly than NPCs, (iii) colonies of hiPSCs would be immediately visually obvious in NPC or neuron cultures, and (iv) NPCs do not show strong expression of critical hiPSC markers such as NANOG, OCT4 or TRA-1-60. Instead, we treat these results as "composition scores" and ignore the scale, focusing on comparing scores for a given cell type across all samples. In this context, the high hiPSC composition score for NPCs likely indicates a "stemness" signal that would be expected to be evident in NPCs and be much lower in neurons.

**Linear mixed model analysis.** The expression variance for each gene was partitioned into the variance attributable to each variable using a linear mixed model implemented in variancePartition v1.5.3[41]. The results were visualized using the package's build-in functions. Categorical variables (i.e., cell type, donor, diagnosis, sex) were modeled as random effects and continuous variables (i.e., cell composition scores) were modeled as fixed effects. Each gene was considered separately and the results for all genes were aggregated afterwards.

**Integration of RNA-Seq data sets**. RNA-Seq data sets were obtained from GTEx (http://www.gtexportal.org), CommonMind (http://www.synapse.org/CMC), BrainSpan (http://www.brainspan.org/), and GEO (https://www.ncbi.nlm.nih.gov/geo/). Gene-level counts for Entrez or HGNC symbols were assigned to the corresponding ENSEMBL identifier. Genes with > 1 count per million in 10% of the samples in each data set were retained. This left 12,670 genes in common across all data sets. All expression values were converted to $\log_2$ RPKM. Quantile normalization was performed on all samples using normalizeBetweenArrays in the limma package[50].

**Concordance analysis**. The correlation between $t$-statistics from differential expression analysis of SZ donors compared to controls in hiPSC-NPCs and hiPSC-neurons in the current analysis compared to differential expression $t$-statistics from five psychiatric diseases[53] and nine cancer types (ref. [54] and Broad Institute TCGA Genome Data Analysis Center (2016): Analysis-ready standardized TCGA data from Broad GDAC Firehose 2016_01_28 run. Broad Institute of MIT and Harvard. Data set. https://doi.org/10.7908/C11G0KM9). Only cancers with at least 30 RNA-Seq experiments were considered.

**Multidimensional scaling**. Analysis was performed using cmdscale function in R based on the distance matrix computed from the pairwise correlation matrix based on all genes in the merged data set.

**Hierarchical clustering**. Hierarchical clustering was implemented in R using complete linkage clustering. A pairwise distance matrix was computed for all samples, and the median distance between all samples in each category were used to create a summary distance matrix using to perform the final clustering.

**Principal components analysis**. PCA was performed on the $\log_2$ CPM values for the each cell type separately, the combined NPC+neuron data set, and the residuals from the combined NPC+neuron data set after removing the effect cell type composition scores.

**Removing effects of heterogeneity in cell type composition**. The gene expression data were adjusted for heterogeneity in cell type composition using a linear model by including the cell type composition score as a covariate. Residuals computed using fibroblast and MEF CTC-scores were used in principal components analysis.

**eQTL enrichment analysis**. The overlap between eQTL genes from the CommonMind Consortium[21] and genes exceeding a variance percentage cutoff for a particular variable in the current analysis is computed. This overlap is then compared to the overlap computed from randomly permutated variance percentages. Each gene is assigned a value based on the percentage of variance explained by a particular variable in the variancePartition analysis. At each of 40 cutoff values, the overlap between genes with values exceeding this cutoff and the 2000 genes with the smallest $p$-values from cis-eQTL analysis is evaluated. The overlap was computed for the observed data and 10,000 data sets with the variance percentages randomly permutated. At each cutoff where > 100 genes are represented, the fold enrichment is computed as

$$\text{fold enrichment} = \frac{\text{overlap}_{\text{observed}}}{\text{overlap}_{\text{permutated}}}.$$

The mean enrichment value and the 90% confidence interval are shown in the plot. Since the most genes (80%) have genome-wide significant eQTLs in the CommonMind data set due to the large sample size, only a set of top genes were considered for enrichment. The top 2000 genes were used here, but the results are not sensitive to varying this number as long as ≤ 10,000 genes are used. Permutation and overlap calculations were performed using regioneR[81].

**Differential expression analysis**. Differential expression analysis was performed with limma/voom v3.28.17[49,50] using duplicateCorrelation to account for measuring multiple samples per donor. Hypothesis testing was performed using the Empirical Bayes procedure in limma. Analysis was corrected for multiple testing using qvalue[82]. Standard error of the $\log_2$ fold change estimates were computed by dividing the $\log_2$ fold change by the moderated t-statistic. Analysis included sex and the cell type composition scores as described above.

**Evaluation of gene set enrichment**. Standard gene set enrichment tests was performed with a hypergeometric test using gene sets from MSigDB[83], MAGMA[84] and additional sets from Fromer et al.[21]

Due to the polygenic nature of COS and the lower power of this study due to its relatively small sample size, changes in gene expression are expected to by subtle and distributed across many genes. The differential expression analysis between SZ and controls did not produce strong results, so we performed extensive enrichment analysis downstream in order extract biological insight. The simplest analysis uses a hard cutoff and considers only genes that pass a given FDR threshold. Genes with FDR < 30% in either cell type were tested with EnrichR[85].

Alternatively, more powerful enrichment analyses do not use a cutoff but instead consider the $t$-statistics of a differential expression test. These tests evaluate enrichment based on genes that are not genome-wide significant, and identify sets of genes for which the distribution of $t$-statistics differs from expectation. Moreover, these tests can use empirical permutations to address the multiple testing problem and determine the significance of gene set enrichments. This permutation approach increases power in small sample sizes with complex correlation structure between genes compared to the standard statistical methods for differential expression and multiple testing correction. This family of tests is well suited to a study of polygenic disease in an underpowered data set. These methods are available in the limma package[50] and work directly on the result of a standard voom analysis[49].

ROAST is a self-contained test that evaluates whether the $t$-statistics of genes in a given set are higher, lower or deviate from zero in either direction more than expected. ROMER is similar to GSEA[83], but uses a sophisticated permutation approach within a linear model framework to increase power.

We modified the standard R code for these methods in order to enable parallelized analysis on a multicore machine and increase the number of permutations. This allows us to run 10,000 permutations for ROAST and 100,000 permutations for ROMER.

**Coexpression analysis**. Analysis was performed with WGCNA[48] on the $\log_2$ CPM values for each cell type. NPC and forebrain neuron samples were analyzed separately and the results were combined downstream. Following standard procedure to ensure an approximately scale-free network, pairwise correlation matrices were raised to a power 9 for both cell types. Topological overlap matrices were computed for each cell type. Coexpression modules were identified with average linkage clustering followed by dynamic brank pruning using the cutree-Dynamic function using the "tree" method with a minimum module size of 20 genes. Enrichment tests for gene sets in each coexpression model were performed with a hypergeometric test.

**Data availability**. All hiPSCs have already been deposited at the Rutgers University Cell and DNA Repository (study 160; http://www.nimhstemcells.org/). RNA-Seq data and reproducible scripts are available at www.synapse.org/hiPSC_COS, as well as GSE106589. Owing to constraints reflecting the original patient consents, the raw RNA-Seq data will be made available by the authors upon reasonable request and IRB approval.

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

## Acknowledgements

K.J.B. is a New York Stem Cell Foundation—Robertson Investigator. This work was partially supported by National Institute of Health (NIH) grants R01 MH101454 (K.J.B.), R01 MH106056 (K.J.B. and P.S.), R01 MH109897 (P.S.) and F31 MH112285 (E.F.), a Brain and Behavior Young Investigator Grant (K.J.B.), and the New York Stem Cell Foundation (K.J.B.). We thank the FACS core at Icahn School of Medicine at Mount Sinai. This work was supported in part through the computational resources and staff expertize provided by Scientific Computing at the Icahn School of Medicine at Mount Sinai. Thanks to Gang Fang, Laura Huckins, Noam Beckmann and David Panchision for critical reading of the manuscript. Jamie Simon drew the original illustrations used in the schematic shown in Fig. 1b. Data were generated as part of the CommonMind Consortium. The CommonMind Consortium includes: Menachem Fromer, Panos Roussos, Solveig K. Sieberts, Jessica S Johnson, Douglas M. Ruderfer, Hardik R. Shah, Lambertus L. Klei, Kristen K. Dang, Thanneer M. Perumal, Benjamin A. Logsdon, Milind C. Mahajan, Lara M. Mangravite, Hiroyoshi Toyoshiba, Raquel E. Gur, Chang-Gyu Hahn, Eric Schadt, David A. Lewis, Vahram Haroutunian, Mette A. Peters, Barbara K. Lipska, Joseph D. Buxbaum, Keisuke Hirai, Enrico Domenici, Bernie Devlin, Pamela Sklar. Funding for the CommonMind Consortium was provided from Takeda Pharmaceuticals Company Limited, F. Hoffman-La Roche Ltd and NIH grants R01MH085542, R01MH093725, P50MH066392, P50MH080405, R01MH097276, RO1-MH-075916, P50M096891, P50MH084053S1, R37MH057881 an R37MH057881S1, HHSN271201300031C, AG02219, AG05138 and MH06692. Brain tissue for the study was obtained from the following brain bank collections: the Mount Sinai NIH Brain and Tissue Repository, the University of Pennsylvania Alzheimer's Disease Core Center, the University of Pittsburgh NeuroBioBank and Brain and Tissue Repositories and the NIMH Human Brain Collection Core. CMC Leadership: Pamela Sklar, Joseph Buxbaum (Icahn School of Medicine at Mount Sinai), Bernie Devlin, David Lewis (University of Pittsburgh), Raquel Gur, Chang-Gyu Hahn (University of Pennsylvania), Keisuke Hirai, Hiroyoshi Toyoshiba (Takeda Pharmaceuticals Company Limited), Enrico Domenici, Laurent Essioux (F. Hoffman-La Roche Ltd), Lara Mangravite, Mette Peters (Sage Bionetworks), Thomas Lehner, Barbara Lipska (NIMH).

## Author contributions

K.J.B., B.J.H., G.E.H., P.S. contributed to experimental design. K.J.B., B.J.H., I.L. completed all cell culture experiments. E.F. conducted microscopy experiments. P.G. and J.R. developed the cohort. D.R. and E.A.S. analyzed genetic data. G.E.H. performed RNA-Seq analysis. K.J.B. and G.E.H. wrote the manuscript.

## Additional information

**Competing interests:** The authors declare no competing financial interests.

