## [Peer Review File · Nature Communications]

Reviewers' comments:

Reviewer #1 (Remarks to the Author):

This study by Hoffman et al. is a tour de force that defines the resolution with which iPSC-based studies can be used to interrogate the genetics of schizophrenia, at least with this paradigm. The work is careful, thoughtful and thorough, and guides the design of future studies that will build on this work and help push iPSC-based studies on schizophrenia. Generally speaking, I am enthusiastic about this manuscript and have few comments/concerns about its contents.

One technical suggestion for future work involves the clonality of reprogramming. The authors picked multiple iPSC clones from each person to define the variation in reprogramming. They are careful to call each clone "presumably clonal" (line 116), a fair characterization since one cannot know guarantee clonality if cells are transduced in bulk before replating. Presumed clones could be siblings derived from the same event before replating. One modification they could use in the future is to split the transduction event into smaller parallel pools: clonality - or at least independent reprogramming events - can then be assured due to the physical isolation of each transduction pool.

The authors discovered that Sendai vectors can be retained in some clones, and they also find that MYC in particular can be selectively retained. This is known and has been described in the past. That they find it independently is reassuring, but they should dig into the literature to buttress their argument and properly cite previous work consistent with their findings. It is a technical detail often buried in papers, so perhaps they could contact the distributor to find the proper citations.

Line 260 "so that expression differences between these cell types are driven by changes in expression magnitude rather than activation of entirely different transcriptional modules" This is one interpretation. Another interpretation is that the NPC population contains neurons, and the neuron population contains NPC, so that transcriptional signatures of both cells are present in both populations. Only purification of each cell type could resolve the difference enough to validate this observation. In fact, I think their CTC variation is more consistent with my alternative interpretation.

Line 297="The "fibroblast" signature. In my opinion, this signature is most likely neural crest or mesenchymal progenitors, two closely related cell types that share features with fibroblasts. The authors could look for CD73 (mesenchymal) or CD271 (neural crest) in a few quick flow experiments to see if they're present in their preparations. They are generally known to contaminate such preparations (see Yuan et al. 2010 for example).

Small details:

Line 118=florescent >fluorescent

Line 161=lincRNA>lncRNA

Line 243=publically>publicly

Define CTC for reader the first time (line 274). ...extensive cell-to-cell (CTC) variation

Kudos to the lab for openly sharing lines and sequencing data: this work is a fantastic advance to the field.

Reviewer #2 (Remarks to the Author):

This manuscript (NCOMMS-17-20479) by Brennand and colleagues reports on the transcriptional signatures of a large set of childhood onset schizophrenia (SZ) cases versus control subject derived hiPSC NPCs and neurons. The large sample set includes various types of replicates allowing an exceptionally thorough analysis of variance in such experiments. This work is of broad interest (beyond the SZ research community) as the approaches utilized and analysis of sources of variance is applicable to any hiPSC based transcriptional research study. One particularly, but peripheral, finding that is of broad importance is the finding of persistent sendai virus genome expression in a significant subset of hiPSC derived cells. As such, this publication has the potential to have a strong and broad impact on this wider research community. Indeed, I was quite excited to see that such work and look forward to seeing it published. That being said a few concerns were identified, that should be addressed to enhance the value of the study to the research community, as well as clarify the interpretation and conclusions of the study in the context of SZ.

- This reviewer's interpretation of Figs 3A-3C, would suggest that in Fig 3C that neurons have a higher (e.g. stronger) 'fibroblast' score. However, the authors conclude a higher fibroblast score for NPCs. It appears that the neurons have more reds and green, while NPCs are more blue and grey - thus to this reviewer neurons have a higher 'fibroblast' score. If this is not accurate, then the text description and detailing of this data needs to be made more clear to the reader.

Likewise, describing a tendency in this data (Figs 3A-3C towards higher scores) is not as robust as could be done. Given the large sample size, it seems reasonable to simply compare whether the NPCs and neurons have a significant difference in their scores or not. A graph is not necessary, but stating whether these scores were significantly different and the magnitude of the difference (and range) would be valuable to readers seeking to apply a similar approach in their own systems. Further addition of all individual CTC scores to Table 2 metadata for each differentiation RNAseq data - would permit the interested reader to compare scores between and within subjects.

- The observation that the authors SZ differentially expressed genes are predominantly not in 'co-expression' modules is rather noteworthy. Given the degree of concordance though with the common mind data, it is perhaps unexpected that this data does not show this same enrichment to the "grey" module. Given the importance of concordance finding with both common mind and NIMH datasets, it would be beneficial to add the NIMH dataset to the co-expression analysis - and then consider the implications of any similarities and differences.

- It is a reasonable hypothesis that SZ relevant biology is not present in the hiPSC-NPC or neuron model systems, that it may require a different cell state, or neuronal connectivity state, to bring out. This should be more clearly discussed and considered as a possible alternative explanation for the failure to detect biologically coherent SZ-associated processes.
- The concordance of the current dataset with CM and HBSS (NIMH) datasets is an important finding. But how specific is this concordance. Comparison of concordance with a few other neurological disorders with comparable statistical power/sample sizes would be quite useful to evaluate if this concordance is specific to SZ, general psychiatric disorders, or merely a neurological disease risk coherence.

Reviewers' comments:

Reviewer #1 (Remarks to the Author):

This study by Hoffman et al. is a tour de force that defines the resolution with which iPSC-based studies can be used to interrogate the genetics of schizophrenia, at least with this paradigm. The work is careful, thoughtful and thorough, and guides the design of future studies that will build on this work and help push iPSC-based studies on schizophrenia. Generally speaking, I am enthusiastic about this manuscript and have few comments/concerns about its contents.

We thank the reviewer for their encouraging comments!

One technical suggestion for future work involves the clonality of reprogramming. The authors picked multiple iPSC clones from each person to define the variation in reprogramming. They are careful to call each clone "presumably clonal" (line 116), a fair characterization since one cannot know guarantee clonality if cells are transduced in bulk before replating. Presumed clones could be siblings derived from the same event before replating. One modification they could use in the future is to split the transduction event into smaller parallel pools: clonality - or at least independent reprogramming events -can then be assured due to the physical isolation of each transduction pool.

We completely agree with the reviewer and appreciate the helpful suggestion. With any luck, the next large cohort to be reprogrammed will be done not just across much smaller transduction pools, but through automated methods!

The authors discovered that Sendai vectors can be retained in some clones, and they also find that MYC in particular can be selectively retained. This is known and has been described in the past. That they find it independently is reassuring, but they should dig into the literature to buttress their argument and properly cite previous work consistent with their findings. It is a technical detail often buried in papers, so perhaps they could contact the distributor to find the proper citations.

We were unaware of this published work and thank the reviewer for pointing this out. We now acknowledge these findings as follows in the text:

"The persistent expression of exogenous reprogramming factors, particularly c-MYC, despite the use of sendai viral non-integrative methods has been previously reported^{89,90} and may reflect the variation of vector replication between cell lines as well as a potential growth advantage of c-MYC expressing cells⁹⁰. Although standard non-integrative methodologies rely upon passive and inefficient omission for the loss of sendai viral

vectors⁹⁰, new methods, such as auto-erasable Sendai virus vectors⁹¹, should facilitate the generation of truly transgene-free hiPSCs.”

Line 260 “so that expression differences between these cell types are driven by changes in expression magnitude rather than activation of entirely different transcriptional modules” This is one interpretation. Another interpretation is that the NPC population contains neurons, and the neuron population contains NPC, so that transcriptional signatures of both cells are present in both populations. Only purification of each cell type could resolve the difference enough to validate this observation. In fact, I think their CTC variation is more consistent with my alternative interpretation.

We thank the reviewer for suggesting this alternative explanation. We have clarified our interpretation in the text as follows:

“Genome-wide, hiPSC-NPCs and hiPSC-neurons express a common set of genes, so that expression differences between these cell types **appear as** changes in expression magnitude rather than activation of entirely different transcriptional modules (**SI Fig. 6**). **Yet this observation is also consistent with continuous variation in CTC whereby the transcriptional signature of each cell type is present in each population at varying levels.**”

Line 297=The “fibroblast” signature. In my opinion, this signature is most likely neural crest or mesenchymal progenitors, two closely related cell types that share features with fibroblasts. The authors could look for CD73 (mesenchymal) or CD271 (neural crest) in a few quick flow experiments to see if they’re present in their preparations. They are generally known to contaminate such preparations (see Yuan et al. 2010 for example).

We completely agree with the reviewer that our observed “fibroblast” signature need not indicate that any actual fibroblasts be present in our NPC or neuron cultures. In fact, the term fibroblast only reflects the source of the single cell RNAseq data: fibroblasts prior to *NGN2*-induction (fibroblast2) or antibiotic-selected fibroblasts following transduction that did not acquire a neuronal signature (fibroblast1). Given that we do not have access to single cell datasets from mesenchymal or neural crest cells, we cannot rule out the obvious possibility that shared gene expression between these three cell types has resulted in us misrepresenting what is in fact a mesenchymal or neural crest cell signature instead as a fibroblast signature.

To do this, we assembled a list of selected mesenchymal (*NT5E* (CD73), *VIM*, *THBS1*, *CDH2*, *FN1*, *ENG*, *ITGB1*, *CD44*, *THY1*) and neural crest (*NGFR* (CD271), *TFAP2A*, *NR2F1*, *NR2F2*, *TWIST1*, *SNAI1*, *SNAI2*, *RARA*, *ALX3*, *ALX4*, *PAX3*, *SOX9*, *SOX10*, *MYC*, *SEMA3A*, *NOTCH1*, *NOTCH2*, *ASCL1*, *CHD7*, *FOXD3*, *NGN1*, *NGN2*, *NGN3*, *NUMB*, *VIM*,

BMP4, *BMP7*) markers. The resulting analyses have been compiled into new **SI Figure 10**, which plots the expression of selected mesenchymal or neural crest cell genes in our hiPSC-NPC and hiPSC-neuron RNA-seq data as well as the CTC reference signatures. With specific reference to CD73 (*NT5E*) and CD271 (*NGFR*), we note that our hiPSC-NPCs and neurons express more *NGFR* than found in any signature except hiPSC, but that *NT5E* expression is shared between hiPSC-NPCs, hiPSC-neurons and fibroblast1.

In order to clarify that the fibroblast1 and fibroblast2 signatures are serving only as a tool to identify unknown non-neural gene expression signatures not represented in the single cell datasets, and could very well imply the presence of other cell type contaminants in the fibroblast and/or hiPSC-derived neural populations that share gene expression similarities with fibroblasts such as mesenchymal cells and/or neural crest cells, we have revised the text as follows:

“Not only is there significant overlap between fibroblast, mesenchymal and neural crest gene expression signatures (reviewed⁵⁴), but both skin fibroblast preparations⁵⁵ and hiPSC-derived NPCs⁵⁶⁻⁵⁸ show evidence of mesenchymal and/or neural crest contaminants. Therefore, it is important to consider the fibroblast₁ and fibroblast₂ signatures only as a tool with which to assess the variability in differentiation quality; high values for the “fibroblast signature” may well imply the presence of non-fibroblast contaminant(s) such as neural crest and/or mesenchymal cells. **SI Fig. 10** plots the expression of key neural crest^{59,60} and mesenchymal⁶¹ genes in our hiPSC-NPC and hiPSC-neuron datasets, as well as the reference panels.”

A

Mesenchymal

B

Neural crest

SI Figure 10. Expression of mesenchymal (top) and neural crest (bottom) markers in hiPSC-NPCs, hiPSC-neurons and across our cell type composition (CTC) signatures.

Plots of selected mesenchymal (*NT5E* (CD73), *VIM*, *THBS1*, *CDH2*, *VTN*, *FN1*, *ENG*, *ITGB1*, *CD44*, *THY1*)⁶¹; <http://www.abcam.com/human-mesenchymal-stromal-cell-marker-panel-cd44-cd45-cd90-cd29-and-cd105-ab93758.html>) and neural crest (*NGFR* (CD271), *TFAP2A*, *NR2F1*, *NR2F2*, *TWIST1*, *SNAI1*, *SNAI2*, *RARA*, *ALX3*, *ALX4*, *PAX3*, *SOX9*, *SOX10*, *MYC*, *SEMA3A*, *NOTCH1*, *NOTCH2*, *ASCL1*, *CHD7*, *FOXD3*, *NGN1*, *NGN2*, *NGN3*, *NUMB*, *VIM*, *BMP4*, *BMP7*) markers^{59,60}; <https://www.rndsystems.com/research-area/neural-crest-cell-markers>) markers in our hiPSC-NPC and hiPSC-neuron RNA-seq data as well as the CTC reference signatures.

Small details:

Line 118=florescent >fluorescent

Line 161=lincRNA>lncRNA

Line 243=publically>publicly

We thank the reviewer for their careful reading and have fixed these typos.

Define CTC for reader the first time (line 274). ...extensive cell-to-cell (CTC) variation

CTC (cell-type composition) is first defined on line 172

Kudos to the lab for openly sharing lines and sequencing data: this work is a fantastic advance to the field.

We thank the reviewer for the helpful comments that have clarified our interpretations.

Reviewer #2 (Remarks to the Author):

This manuscript (NCOMMS-17-20479) by Brennand and colleagues reports on the transcriptional signatures of a large set of childhood onset schizophrenia (SZ) cases versus control subject derived hiPSC NPCs and neurons. The large sample set includes various types of replicates allowing an exceptionally thorough analysis of variance in such experiments. This work is of broad interest (beyond the SZ research community) as the approaches utilized and analysis of sources of variance is applicable to any hiPSC based transcriptional research study. One particularly, but peripheral, finding that is of broad importance is the finding of persistent sendai virus genome expression in a significant subset of hiPSC derived cells. As such, this publication has the potential to have a strong and broad impact on this wider research community. Indeed, I was quite excited to see that such work and look forward to seeing it published. That being said a few concerns were identified, that should be addressed to enhance the value of the study to the research community, as well as clarify the interpretation and conclusions of the study in the context of SZ.

This reviewer's interpretation of Figs 3A-3C, would suggest that in Fig 3C that neurons have a higher (e.g. stronger) 'fibroblast' score. However, the authors conclude a higher fibroblast score for NPCs. It appears that the neurons have more reds and green, while NPCs are more blue and grey - thus to this reviewer neurons have a higher 'fibroblast' score. If this is not accurate, then the text description and detailing of this data needs to be made more clear to the reader.

We thank the reviewer for noticing this discrepancy. We have corrected this error and the text now reads:

“Unexpectedly, hiPSC-neurons had a higher fibroblast₁ score...”

Likewise, describing a tendency in this data (Figs 3A-3C towards higher scores) is not as robust as could be done. Given the large sample size, it seems reasonable to simply compare whether the NPCs and neurons have a significant difference in their scores or not. A graph is not necessary, but stating whether these scores were significantly different and the magnitude of the difference (and range) would be valuable to readers seeking to apply a similar approach in their own systems. Further addition of all individual CTC scores to Table 2 metadata for each differentiation RNAseq data - would permit the interested reader to compare scores between and within subjects.

We thank the reviewer for this suggestion. We now include effect sizes and p-values in the text. Here we report p-values from a linear model, but using the Wilcoxon test gives similar results.

“As expected, hiPSC-neuron samples had a higher neuron CTC score than hiPSC-NPCs (mean increase = 0.06, $p < 1.05e-6$ by linear model)

(Fig. 3A), while hiPSC-NPCs had a higher hiPSC CTC score (mean increase = 0.20, $p < 1.49 \times 10^{-31}$ by linear model), consistent with a “stemness” signal (a neural stem cell profile was lacking from our reference) (Fig. 3B). Unexpectedly, hiPSC-neurons had a higher fibroblast₁ score (mean increase = 0.09, $p < 1.1 \times 10^{-6}$ by linear model) (Fig. 3C).”

We also now include the CTC scores in Supplementary Table 2.

The observation that the authors SZ differentially expressed genes are predominantly not in 'co-expression' modules is rather noteworthy. Given the degree of concordance though with the common mind data, it is perhaps unexpected that this data does not show this same enrichment to the "grey" module. Given the importance of concordance finding with both common mind and NIMH datasets, it would be beneficial to add the NIMH dataset to the co-expression analysis - and then consider the implications of any similarities and differences.

This is a helpful suggestion. We now add the NIMH HBCC dataset to the co-expression analysis; the results, presented in new Figure 5, are consistent with the lack of enrichment to the “grey” model.”

“Genes identified by genetic studies (i.e. common variants, CNVs, rare loss of function and de novo variants) and case/control signatures from two post mortem datasets (the CommonMind Consortium (CMC)⁴⁸ and the NIMH Human Brain Collection core (HBCC)) showed moderate enrichment in many modules, but did not strongly overlap with the grey modules enriched for differentially expressed genes from this study.”

Figure 5: Clustering of genes into coexpression modules reveals module-specific enrichments. Enrichment significance ($-\log_{10}$ p-values from hypergeometric test) are shown for coexpression modules from hiPSC-NPCs and hiPSC-neurons. Each module is assigned a color and only modules with an enrichment passing the Bonferroni cutoff in at least one category is shown. Enrichments are shown for gene sets from RNA-Seq studies of differential

expression between schizophrenia and controls; genetic studies of schizophrenia, neuronal proteome⁴⁸; and cell composition scores from hiPSC-NPCs and hiPSC-neurons in this study. P-values passing the 5% Bonferroni cutoff are indicated by ‘*’, and p-values less than 0.05 are indicated with ‘.’.

We agree that this is an unexpected finding, and so now specifically draw attention to this:

“Given the degree of concordance in the SZ differentially expressed genes between the hiPSC-NPCs, hiPSC-neurons, CMC and NIMH HBCC datasets (Fig. 6D,E), the lack of enrichment of the CMC or NIMH HBCC differentially expressed genes in the “grey module” of our coexpression analysis (Fig. 5) is noteworthy. Although the concordance and coherence of the signal between hiPSC-NPCs and hiPSC-neurons with two post mortem datasets was relatively low due, we believe this reflects the small sample size and low power of our current study and predict that both will increase with expanding sample sizes in future studies.”

It is a reasonable hypothesis that SZ relevant biology is not present in the hiPSC-NPC or neuron model systems, that it may require a different cell state, or neuronal connectivity state, to bring out. This should be more clearly discussed and considered as a possible alternative explanation for the failure to detect biologically coherent SZ-associated processes.

The reviewer is of course correct that SZ relevant biology need not be present in simple monolayer hiPSC-NPC and hiPSC-neuron populations. We now explicitly consider this in the discussion as follows:

“Yet this shared biology did not yield enrichments at the pathway or network level in the diagnosis-dependent differentially expressed genes observed between hiPSC-NPCs and hiPSC-neurons with either post mortem dataset. Moving forward, increasing the sample size of hiPSC-based cohorts may improve this concordance and biological coherence. Alternatively, it is possible that many SZ-associated processes are not present in simple monolayer hiPSC-NPC and hiPSC-neuron populations; relevant aspects of SZ biology may only be detected through activity-dependent processes arising from complex neuronal circuitry, following oligodendrocyte myelination, astrocyte support or microglia pruning, or after exposure to neuroinflammation or environmental stimuli. While the best strategy to improve biological significance is to strive to enhance the complexity of hiPSC-based models, the surest approach to improve the power of case-control comparisons is to integrate a growing number of post mortem and hiPSC studies.”

The concordance of the current dataset with CM and HBSS (NIMH) datasets is an important finding. But how specific is this concordance. Comparison of

concordance with a few other neurological disorders with comparable statistical power/sample sizes would be quite useful to evaluate if this concordance is specific to SZ, general psychiatric disorders, or merely a neurological disease risk coherence.

This is an excellent idea. We now include an additional plot (Figure 6F) showing the strong concordance between both our SZ NPC (green) and neuron (orange) DE with ASD and BD datasets but not alcoholism, MDD or a variety of cancer types. We note this new analysis in the text as follows:

“To a lesser extent, this concordance was also detected in ASD and BD post-mortem datasets, but not in other neuropsychiatric disorders such as alcoholism and major depression disorder, or a variety of cancer types (Fig. 6F), indicating the specificity of our results.”

Figure 6: Differential expression between schizophrenia and controls. A,B) Volcano plot showing \log_2 fold change between cases and controls and the $-\log_{10} p$ -value for each gene in **A)** hiPSC-NPC and **B)** hiPSC-neuron samples. Genes are colored based on false discovery rate: light red (FDR < 10%), dark red (FDR < 30%), grey (n.s.: not significant). Names are shown for genes with FDR 30%. Dotted grey line indicates Bonferroni cutoff corresponding to a p-value of 0.30. Dashed dark red line indicates FDR cutoff of 30% computed by qvalue (Storey, 2002). **C)** \log_2 fold change between cases and control in hiPSC-NPCs (x-axis) compared to \log_2 fold change between cases and controls in hiPSC-neurons (y-axis). Genes are colored according to differential expression results

from combined analysis of both cell types: light red (FDR < 10%), dark red (FDR < 30%), grey (n.s.: not significant). Error bars represent 1 standard deviation around the \log_2 fold change estimates. **D,E**) Analysis of concordance between differential expression results of schizophrenia versus controls from the current study and two adult post mortem cohorts⁴⁸. Concordance is evaluated based on spearman correlation between t-statistics from two datasets. **D**) Spearman correlation between t-statistics from the current study (from hiPSC-NPCs and -neurons) and the two post mortem cohorts. **E**) $-\log_{10}$ p-values from a one-sided hypothesis test for the Spearman correlation coefficients from (**D**) being greater than zero. **F**) Concordance of t-statistics with differential expression results from case-control analysis of five psychiatric diseases⁹² and tumor-normal analysis of nine cancer types⁹³.

The following text was added to the **Methods** to explain the source of the datasets used:

“Concordance Analysis

The correlation between t-statistics from differential expression analysis of SZ donors compared to controls in hiPSC-NPCs and hiPSC-neurons in the current analysis compared to differential expression t-statistics from five psychiatric diseases³³ and nine cancer types (³⁴ and Broad Institute TCGA Genome Data Analysis Center (2016): Analysis-ready standardized TCGA data from Broad GDAC Firehose 2016_01_28 run. Broad Institute of MIT and Harvard. Dataset. <https://doi.org/10.7908/C11G0KM9>). Only cancers with at least 30 RNA-seq experiments were considered.”

REVIEWERS' COMMENTS:

Reviewer #1 (Remarks to the Author):

The authors have addressed all minor concerns I raised in the initial review. This is a manuscript of great importance not only to the SZ community but to the larger PSC/neural-focused stem cell field. Few groups have pushed the analysis to this level of granularity, and they have provided a solid foundation for future studies.

Mark Tomishima

Reviewer #2 (Remarks to the Author):

All my concerns have been addressed - this is a fantastic dataset and analysis that will be an important contribution to the field.